# Super-resolution imaging of RAD51 and DMC1 in DNA repair foci reveals dynamic distribution patterns in meiotic prophase

Johan A. Slotman[1,2‡], Maarten W. Paul[1‡], Fabrizia Carofiglio[3], H. Martijn de Gruiter[1], Tessa Vergroesen[3], Lieke Koornneef[3], Wiggert A. van Cappellen[1], Adriaan B. Houtsmuller[1,2‡], Willy M. Baarends[3‡]*

1 Erasmus Optical Imaging Centre, Department of Pathology, Erasmus MC—University Medical Center, Rotterdam, The Netherlands, 2 Department of Pathology, Erasmus MC—University Medical Center, Rotterdam, The Netherlands, 3 Department of Developmental Biology, Erasmus MC—University Medical Center, Rotterdam, The Netherlands

‡ These authors are joint senior authors on this work. JAS and MWP share first authorship on this work.
* w.baarends@erasmusmc.nl

**Data Availability Statement:** All relevant data are within the manuscript and its Supporting Information files.

## Abstract

The recombinase RAD51, and its meiosis-specific paralog DMC1 localize at DNA double-strand break (DSB) sites in meiotic prophase. While both proteins are required during meiotic prophase, their spatial organization during meiotic DSB repair is not fully understood. Using super-resolution microscopy on mouse spermatocyte nuclei, we aimed to define their relative position at DSB foci, and how these vary in time. We show that a large fraction of meiotic DSB repair foci (38%) consisted of a single RAD51 nanofocus and a single DMC1 nanofocus (D1R1 configuration) that were partially overlapping with each other (average center-center distance around 70 nm). The vast majority of the rest of the foci had a similar large RAD51 and DMC1 nanofocus, but in combination with additional smaller nanofoci (D2R1, D1R2, D2R2, or DxRy configuration) at an average distance of around 250 nm. As prophase progressed, less D1R1 and more D2R1 foci were observed, where the large RAD51 nanofocus in the D2R1 foci elongated and gradually oriented towards the distant small DMC1 nanofocus. D1R2 foci frequency was relatively constant, and the single DMC1 nanofocus did not elongate, but was frequently observed between the two RAD51 nanofoci in early stages. D2R2 foci were rare (<10%) and nearest neighbour analyses also did not reveal cofoci formation between D1R1 foci. However, overall, foci localized nonrandomly along the SC, and the frequency of the distance distributions peaked at 800 nm, indicating interference and/or a preferred distance between two ends of a DSB. DMC1 nanofoci where somewhat further away from the axial or lateral elements of the synaptonemal complex (SC, connecting the chromosomal axes of homologs) compared to RAD51 nanofoci. In the absence of the transverse filament of the SC, early configurations were more prominent, and RAD51 nanofocus elongation occurred only transiently. This in-depth analysis of single cell landscapes of RAD51 and DMC1 accumulation patterns at DSB repair sites at super-resolution revealed the variability of foci composition, and defined functional consensus configurations that change over time.

**Funding:** FC was funded by the Netherlands Organization for Scientific Research (NWO) through the ALW Open Programme 819.02.020. MP was funded by NWO-CW ECHO 104126. The funders had no role in study design, data collection and analysis, decision to publish, or preparation of the manuscript.

**Competing interests:** The authors have declared that no competing interests exist.

## Author summary

Meiosis is a specific type of cell division that is central to sperm and egg formation in sexual reproduction. It forms cells with a single copy of each chromosome, instead of the two copies that are normally present. In meiotic prophase I, homologous chromosomes must connect to each other, to be correctly distributed between the daughter cells. This involves the formation and repair of double-strand breaks in the DNA. Here we used super-resolution microscopy to elucidate the localization patterns of two important DNA repair proteins: RAD51 and DMC1. We found that repair sites most often contain a single large nanofocus of both proteins, with or without one additional smaller nanofocus of either protein. RAD51 protein nanofoci displayed lengthening as meiotic prophase progressed, and localized somewhat closer to the protein axis that mediates the physical connection (synapsis) between homologous chromosomes compared to DMC1 nanofoci. When chromosome synapsis was disturbed, we observed changes in the dynamics of protein accumulation patterns, indicating that they actually correspond to certain repair intermediates changing in relative frequency of occurrence. These analyses of single meiotic DNA repair foci reveal the biological variability in protein accumulation patterns, and the localization of RAD51 and DMC1 relative to each other, thereby contributing to our understanding of the molecular basis of meiotic homologous recombination.

## Introduction

During meiosis, correct homologous chromosome pairing and separation requires the repair of programmed, meiosis-specific, DNA double-strand breaks (DSBs), induced by a meiosis-specific topoisomerase type II-like complex [1–3], in species ranging from yeast to mammals. The machinery that generates and repairs the DSBs is meiosis-specific, but contains many proteins that also function in homologous recombination (HR) repair of DSBs in somatic cells (reviewed in [4]). In somatic HR (mainly active during S or G2 phase), the DNA of DSBs is resected, resulting in the formation of two 3'single-strand (ss) DNA ends, coated by the ssDNA binding protein complex RPA. Subsequently, RPA is replaced by the recombinase RAD51. This enzyme forms a protein filament on the DNA and is capable of mediating strand invasion and strand displacement (D-loop formation) [5]. This allows subsequent steps in repair, involving recovery of the missing information from the intact sister chromatid.

Meiotic DSB ends are also resected, but in addition to RPA, meiosis-specific ssDNA binding proteins also associate with the processed ssDNA ends [6,7]. RPA is then displaced by the canonical recombinase RAD51, and its meiosis-specific paralog DMC1 (53.6% amino acid identity to RAD51 in mouse) [8,9]. The two recombinases appear to colocalize in mouse spermatocytes and oocytes when imaged with standard microscopy techniques [10,11]. In *A. thaliana* atRAD51 and atDMC1 have been detected as paired foci, suggesting that each of the two DSB ends may be coated by a different recombinase [12]. However, recent super-resolution imaging in *S. cerevisiae* has indicated that multiple small DMC1 and RAD51 filaments may accumulate on both ends of a meiotic DSB, and paired co-foci were observed at lower resolution [13]). Mouse spermatocytes are very suitable for immunocytology, due to their relatively large size, and well-organized patterns of chromosomal axes. These are used to substage meiotic prophase, using antibodies targeting meiosis-specific chromosomal axis proteins such as SYCP2 and SYCP3, that form the platform on which the programmed DSBs are processed [14]. Here we addressed the nanoscopic localization of RAD51 and DMC1 during mouse

meiotic prophase. First, we assessed the overall distribution of RAD51/DMC1 foci in the nucleus using confocal microscopy. Next, we employed a combination of Structured Illumination Microscopy (SIM) and direct Stochastic Optical Reconstruction Microscopy (dSTORM) in two colours to visualize nanoscopic details of RAD51 and DMC1 foci in mouse meiotic prophase nuclei. We compared the localization pattern of the two recombinases in wild type spermatocytes with spermatocytes lacking the transverse filament protein SYCP1 ($Sycp1^{-/-}$). In the absence of this core component of the synaptonemal complex homologous chromosomes align but fail to synapse, resulting in the persistence of meiotic DSB repair foci [15].

Our results show that most repair foci contain single RAD51 and DMC1 nanofoci that are in close proximity to each other, with or without one much smaller additional RAD51 or DMC1 nanofocus at larger distance. As prophase progresses, configurations become more complex, and the major domain elongates, but this is dependent on the presence of SYCP1. One of the possible interpretations of these data may be that foci configurations with a single partially overlapping RAD51 and DMC1 structure represent filament formation on one end of a meiotic DSB, and that the distance to the other end is highly variable, precluding frequent observation of co-foci. In addition, the relatively frequent occurrence of the D2R1 and D1R2 configurations indicate that there may be stochastic variations in filament formation and/or in chromatin binding patterns of RAD51 and DMC1. This work is a first step towards unravelling the exact molecular composition of the meiotic recombination machinery in time and space in single cells.

## Results

### Non-random distribution of RAD51-DMC1 foci along axial elements

Previous analyses performed on *S. cerevisiae* meiocytes have indicated non-random occurrence of pairs of RAD51-DMC1 co-foci [13]. RAD51 and DMC1 also colocalize in easily discernible repair foci in mouse spermatocytes and oocytes [8,11] but formation of pairs of such foci has not been described, and is also not immediately evident from the microscopic images that can be obtained (Fig 1A). In mouse, these foci are usually analysed in combination with visualization of the axial/lateral elements of the SC, since it is known that the meiotic DSBs localize along these axes. Previously, non-random distribution of markers of repair foci along the axial elements of specific chromosomes has been shown for late zygotene and pachytene spermatocytes, providing evidence for different levels of crossover interference [16–18], but such analyses have not been performed for earlier stages. To ensure nonbiased quantification of immunosignals we used FIJI to automatically select foci in leptotene and zygotene nuclei, see Materials and Methods for details. We then counted the numbers of RAD51 and DMC1 foci (S1A Fig), and used these numbers to simulate random distributions of the same number of artificially generated foci. We observed that RAD51 and DMC1 foci localization was mostly confined to the areas in which the axial and lateral elements of the SC were forming upon visual inspection of the images. Since this feature generates a non-random organisation of the foci in the nucleus, we used a mask to select only those foci that were located on the chromosomal axes (examples of selected foci and raw images are shown in Fig 1A–1C), and used these foci numbers to also simulate random distributions of the same number of artificially generated foci along the areas covering the SYCP3 signal for each nucleus as described in Materials and Methods (see examples in Fig 1A–1C). This second analysis allowed assessment of random or non-random distribution of DSB-repair foci along the axial/lateral elements. In leptotene, the use of the mask led to a reduction of 40 and 38% of the DMC1 and RAD51 foci, respectively, whereas 27% of DMC1 foci and 22% of the RAD51 foci located outside of the mask in zygotene. These "lost foci" are expected to represent background signal, as well as

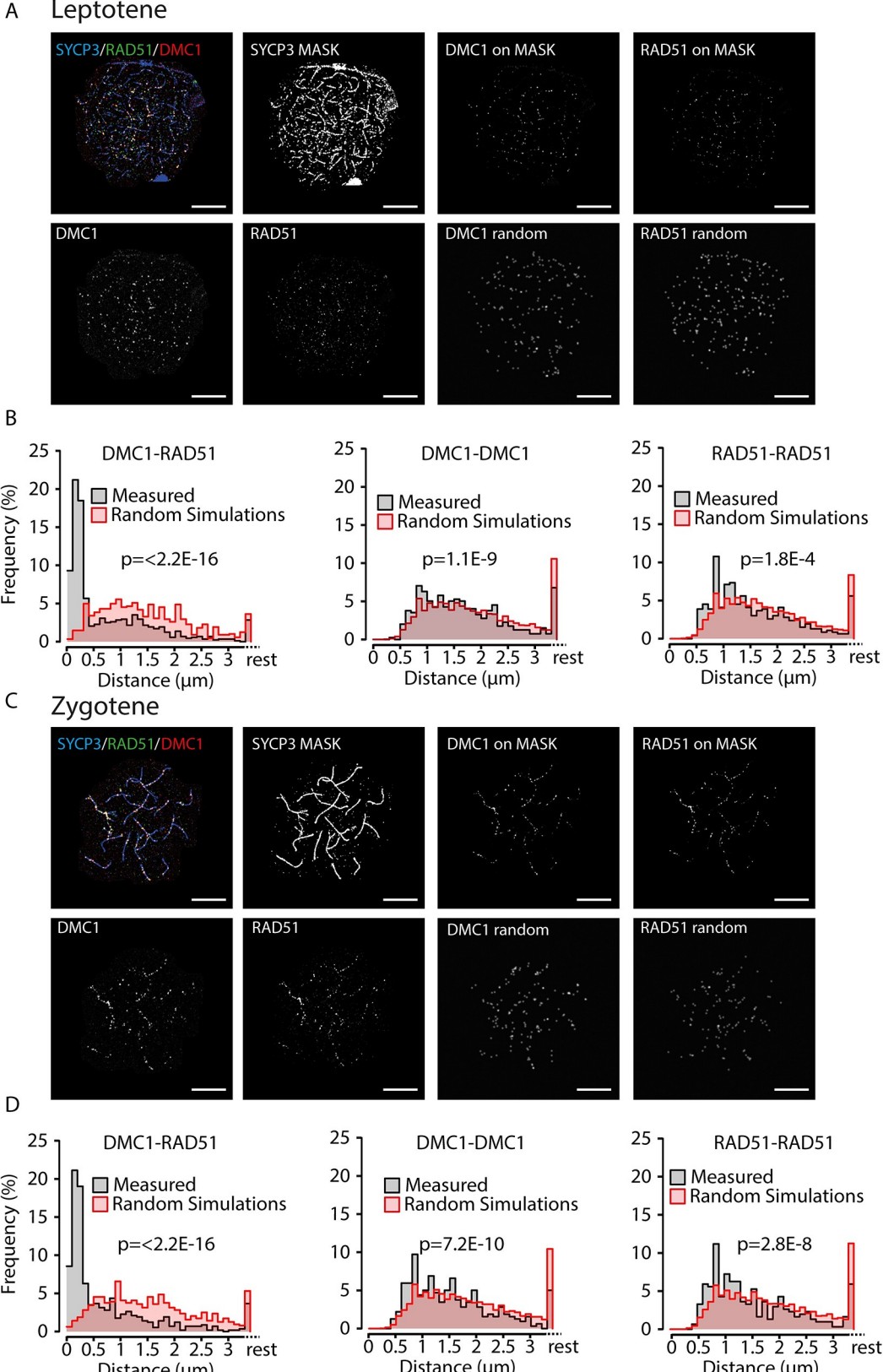

**Fig 1. Nearest neighbour analyses of confocal microscopy images of RAD51 and DMC1 foci on the synaptonemal complex axes.** A), C) Top left, example confocal image of triple stained leptotene (A) and zygotene (C) nucleus, with primary antibodies for RAD51, DMC1, and SYCP3, and appropriate secondary antibodies conjugated with Alexa 488 (green), Alexa 647 (red), and Alexa 555 (blue), respectively; single DMC1 and RAD51 images are shown in greyscale below; the SYCP3 mask generated as described in Materials and Methods is shown to the right of the triple staining; the two top right images show the DMC1 and RAD51 foci that localize on the mask, and below them, the same number of foci randomly distributed on the mask (scale bars represent 10 μm). B), D) Relative frequency distribution of nearest neighbour distances between DMC1 and RAD51 (left), DMC1 and DMC1 (middle), and RAD51 and RAD51 (right) foci on the SYCP3 masked regions in leptotene (B, n = 7 nuclei; 606 DMC1 foci, 712 RAD51 foci) and zygotene (D, n = 6 nuclei; 471 DMC1 foci, 462 RAD51 foci) wild type nuclei. Distances were binned in 100 nm bins, distances larger than 3.4 μm were labelled as rest. Grey bars, experimental data; red bars, simulated data (see Materials and Methods). p-values of Kolmogorov-Smirnov tests are indicated in the graphs.

some sites of true RAD51 and DMC1 accumulation on DSB repair sites not localised on the axial/lateral elements. Subsequently, we determined the nearest distance between RAD51 and DMC1 foci, as well as the RAD51-RAD51 and DMC1-DMC1 distances for both types of foci selection. These analyses showed that 55% (all foci) or 80% (on the mask), and 57% (all foci) or 67% (on the mask) of the analysed DMC1 foci on had a RAD51 neighbour at a distance shorter than 300 nm in leptotene and zygotene, respectively (For p-values and other statistical parameters see S2 Table), reflecting the overall colocalization. Analyses of DMC1-DMC1 and RAD51-RAD51 distances also revealed a non-random distribution (Fig 1B and 1D, S1B Fig and S1C Fig) for both the analyses performed on all foci, and the foci on the mask. Also, for both types of analyses distances between 500 and 800 nm occurred more frequently than expected based on a random distribution. This could be explained by the fact that DSB foci are generally excluded from specific regions, such as constitutive heterochromatin and near centromeric areas, causing foci to be in closer proximity to each other than expected based on random distribution. However, the rather sharp peaks of RAD51-RAD51 and DMC1-DMC1 nearest neighbour distances around 800 nm in zygotene within the DSB-foci positive SC regions, indicate additional non-random distribution. This could reflect a form of interference between DSBs, since the foci appear organized as beads along a string of SC. However, we also cannot exclude that some of these distant foci actually represent a pair of single ends of a DSB at a preferred distance of around 800 nm to each other, similar to the co-foci observed in yeast, at shorter distance [13]. The fact that the peaks around 800 nm are somewhat more pronounced when the mask is not used, may (at least in part be caused by background signal. In addition, some foci may localise just outside the mask, away from the axes, at around 800 nm distance from foci on the axes, and contribute to the 800 nm peak frequency.

## Composition of meiotic recombination foci revealed by super-resolution imaging

To establish precisely how RAD51 and DMC1 accumulate relative to each other at distances smaller than 300 nm, we visualized RAD51, DMC1, and SYCP3, using SIM and dSTORM, (Fig 2A–2E). By utilizing a microscope that combines SIM and dSTORM, we were able to visualise the same field-of-view applying both techniques with the same objective lens (Fig 2A and 2B). The SIM images were used to visualise synaptonemal complexes (SCs), to be able to identify the substage of meiotic prophase and meiotic DSB foci (also in the SIM image), which were further analysed in images acquired by dSTORM. In DMC1 and RAD51 co-staining experiments, the two proteins displayed distinct localisation patterns, both in SIM and dSTORM images (Fig 2C and 2D).

A total dataset of 2315 manually selected foci (ROIs of 600 nm diameter circles drawn around the centre of each focus, to allow separate selection of foci at distances of >300 nm)

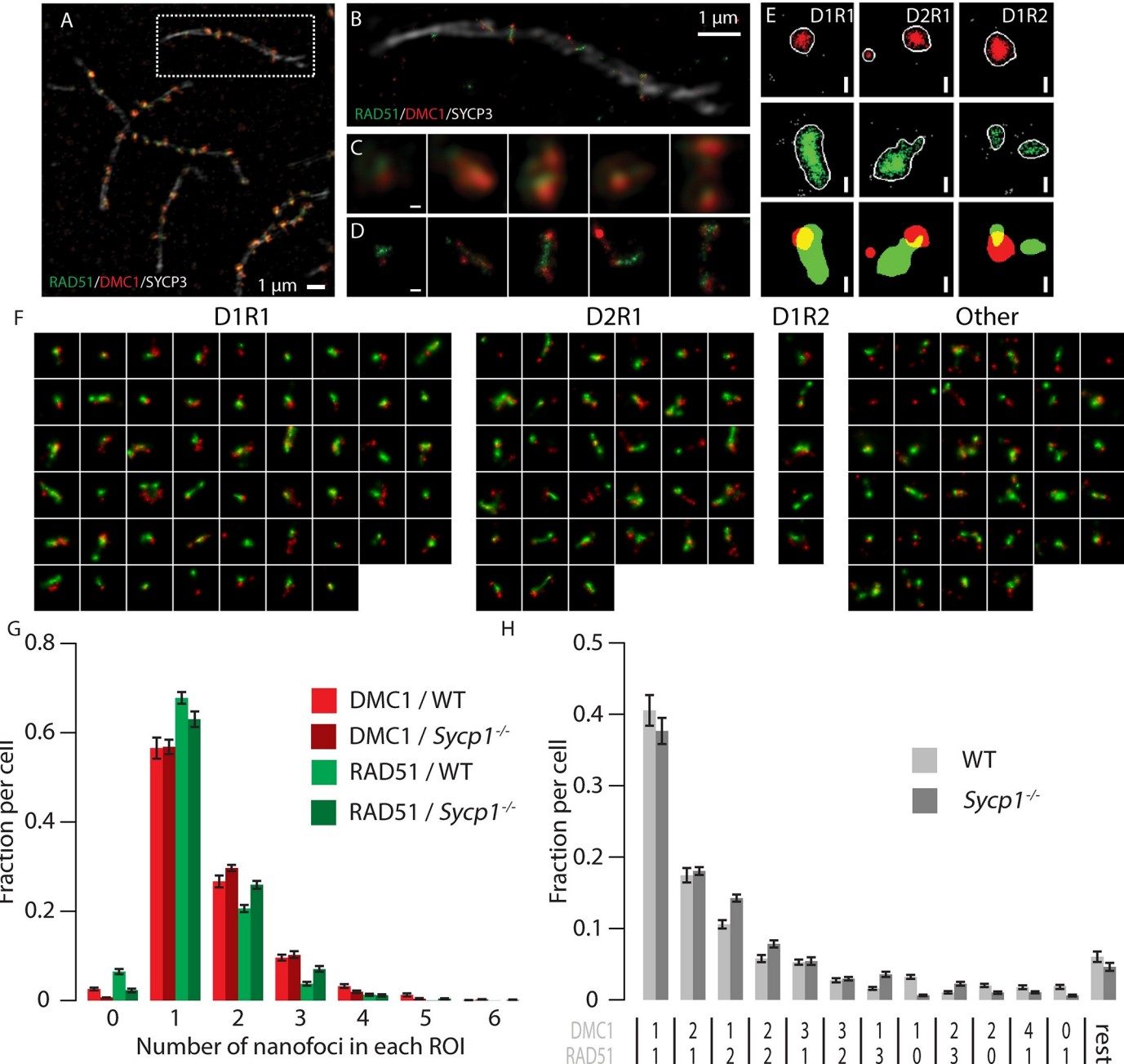

**Fig 2. Meiotic DSB foci in super-resolution.** A) Cropped region from a SIM image of a spread mouse late zygotene nucleus immunostained with primary antibodies for RAD51, DMC1, and SYCP3, and appropriate secondary antibodies conjugated with Alexa 488 (green), Alexa 647 (red), and Alexa 555 (white), respectively. B) SYCP3 SIM overlayed with RAD51/DMC1 dSTORM images of boxed region in A). C) Close-up of single DSB foci present on the synaptonemal complex shown in A). D) The same foci visualized with dSTORM. E) Single DSB foci of three types (left panels D1R1, middle panels D2R1, right panels D1R2) represented by 2 different visualisation/analysis methods: scatter plot of localisations and merged binary representation of the kernel density estimation. F) Compilation of all ROIs of a single late zygotene nucleus (indicated with an asterisk in S2 Fig). ROIs are sorted by their DxRy configuration, from most frequent to rare configuration. G) Fraction of foci containing the indicated number of RAD51 or DMC1 nanofoci per focus as a percentage of the number of foci per genotype. H) fraction of foci containing the indicated combinations of RAD51 and DMC1 nanofoci per focus as a percentage of the number of foci per genotype. Combinations that represented less than 1% of the foci in both wild type and Sycp1-/- were grouped in the category referred to as rest. Scale bars in C-E represent 100 nm. Error bars represent SEM values. Results of statistical analyses are described in S2 Table.

was generated by analysis of 18 nuclei in different meiotic substages, imaged in four independent experiments (S2 Fig, S1 Table). The maximum number of foci per nucleus was observed in early zygotene, corresponding well with what we and others have reported previously [11,19,20].

## Most foci contain a major configuration consisting of one RAD51 and one DMC1 nanofocus

Many different configurations of RAD51 and DMC1 assemblies can be discerned (Fig 2F). To quantify and categorize the different patterns of RAD51 and DMC1 localisation events objectively, we generated binary images to identify specific discrete RAD51 and DMC1 areas wherein localisation event number and density fulfilled the set criteria (see materials and methods), within the ROIs. We refer to these as nanofoci, to discriminate them from the DSB repair foci that are represented by the ROIs (Fig 2E)[21]. We quantified the number of nanofoci within each ROI and observed that for both RAD51 and DMC1 a single nanofocus within a ROI was most frequently observed (Fig 2G). Foci with multiple RAD51 or DMC1 nanofoci were also present, and were somewhat more frequent (1.6 fold, p = 0.0106 (two-sided t-test) for DMC1 compared to RAD51 (Fig 2G, S2 Table). Next, we quantified the different RAD51 and DMC1 nanofocus combinations in our ROIs dataset in order to assess how the two recombinases relate to each other within each ROI. In the distribution of nanofocus combinations, 69% of the total population of ROIs fell within three specific groups: one DMC1 nanofocus and one RAD51 nanofocus (D1R1, 41%), two DMC1 nanofoci with a single RAD51 nanofocus (D2R1, 17%), or two RAD51 nanofoci and one DMC1 nanofocus (D1R2, 11%) (Fig 2H). Only 6% of the foci contained 2 nanofoci of each recombinase (D2R2), and all other combinations occurred at lower frequencies.

We also analysed a mouse mutant model in which assembly of the synaptonemal complex (SC) is incomplete due to the absence of the central or transverse filament of the SC (*Sycp1*[−/−], 2 animals, two independent experiments, 10 nuclei, 2042 manually selected foci (S3 Fig, S1 Table) [15]. In spermatocytes from these mice, homologous chromosomes show pairing but no synapsis, and the distances between paired axial elements are larger than between lateral elements in synapsed SCs in the wild type (around 80 nm in wild type and 200 nm in the knockout) [15]. In this mutant, leptotene appears normal, and the number of DSB foci observed at this stage is similar to the maximum number observed in wild type spermatocytes, but the failure to synapse disturbs subsequent stages, and prevents completion of meiotic DSB repair ([15,22–25] and S3 Fig). Overall, DxRy configurations were present in similar frequencies in wild type and *Sycp1*[−/−] nuclei, although D1R2 and other configurations with more than one RAD51 nanofocus were observed somewhat more frequently (1.35 fold, p = 0.048 (two-sided t-test)) in the knockout (Fig 2G and 2H and S2 Table).

Next, we also manually classified all binary images based on the observed shapes and sizes of the binary images of the nanofoci. We observed several typical configurations. First, we defined a so-called "simple" structure consisting of a relatively large D nanofocus and a large R nanofocus with roundish shapes, that partially overlapped, and that may or may not have additional RAD51 or DMC1 nanofoci that were separate and generally much smaller (Fig 3A: "simple"). This simple structure was frequently observed in wild type as well as *Sycp1*[−/−] spermatocytes (Fig 3B–3E). Interestingly, its relative frequency was rather constant in wild type, but showed a tendency to decrease in zygotene- and pachytene-like nuclei compared to leptotene in *Sycp1*[−/−] spermatocytes (2.3 fold difference, p = 0.06, Fig 3D and 3E).

Second, a combination of more complex partially overlapping shapes of a major D and major R nanofocus was also frequently observed in both wild type and *Sycp1*[−/−] spermatocytes (Fig 3A:

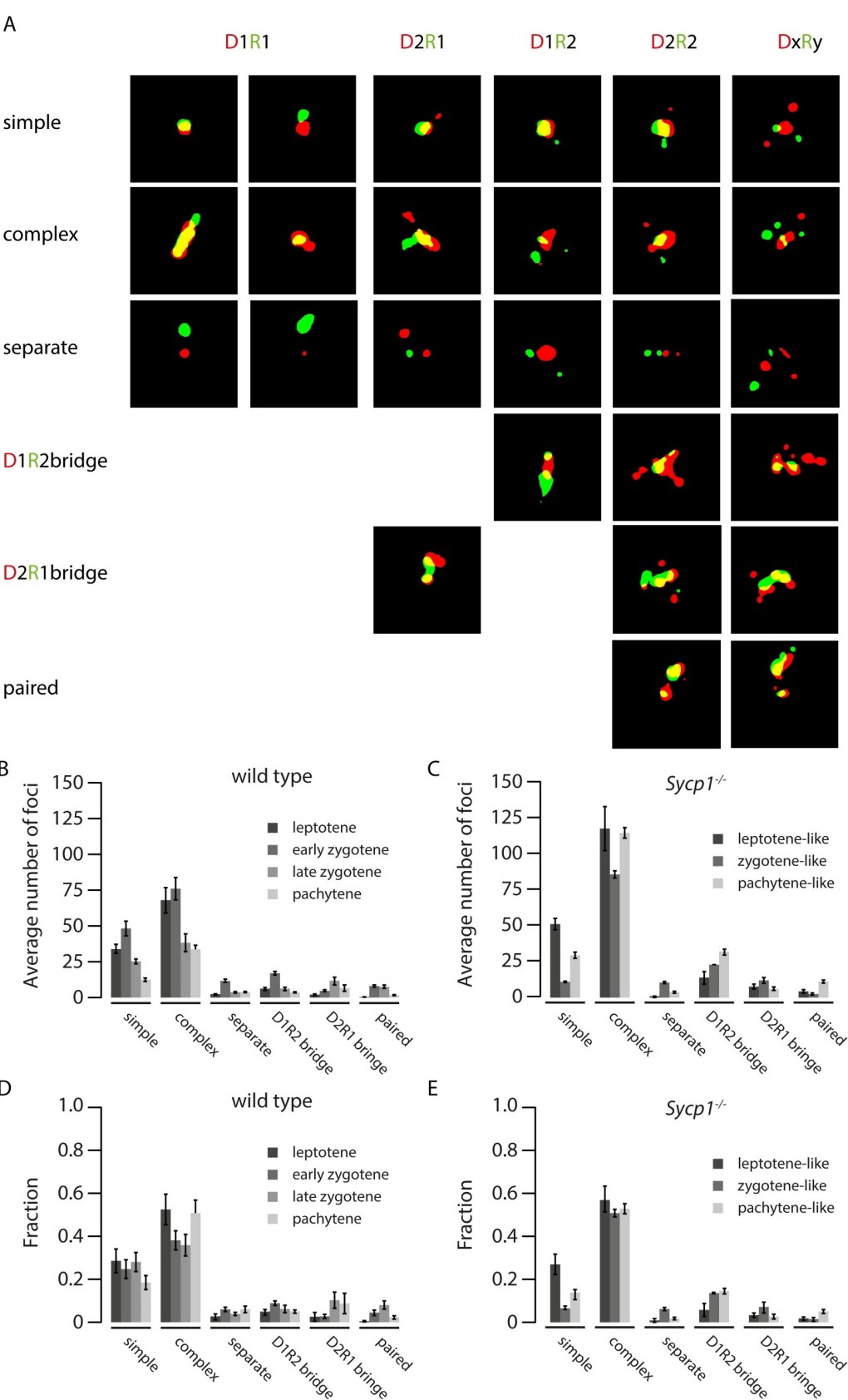

**Fig 3. Morphological classification of RAD51-DMC1 configurations.** A). All foci were classified as simple, complex, separate, D1R2 bridge, D2R1 bridge, or paired as described in the main text. Examples of each are shown for (from left to right), D1R1 D1R2, D2R1, D2R2, DxRy. B) and C) Average number of foci per cell, per morphological classification, and D) and E) average fraction of foci per cell per morphological classification in leptotene (dark grey), zygotene (gray) and pachytene (light gray) of wild type (B, D) and Sycp1$^{-/-}$ (C, E) nuclei. Error bars indicate SEM.

"complex", and Fig 3B–3E). Again, additional nanofoci were usually relatively small compared to the two main nanofoci. Together, these so-called simple and complex foci comprised the majority of all configurations in both wild type (69±2.8% (SEM)) and Sycp1$^{-/-}$ nuclei (70±3.6% (SEM)). This result indicated that the D1R1 foci could actually be representative for a much larger fraction of the DxRy foci if the small additional nanofoci were considered "satellites".

A third notable structure that was observed for D1R2, D2R1, and also for ROIs containing more nanofoci, was termed "bridge" (Fig 3A "bridges", 11% of all foci in wild type and 14% in Sycp1$^{-/-}$). These contained 2 DMC1 nanofoci that were connected (via a partial overlap of signal) by one RAD51 nanofocus (D2R1 bridge), or the reverse situation (D1R2 bridge), with or without additional nanofoci. For the D2R1 bridge structure, the relative frequency tended to be higher in late zygotene and pachytene compared to the earlier stages in wild type (3.5 fold increase, p = 0.056), but not in the Sycp1$^{-/-}$ spermatocytes (Fig 3D and 3E). Conversely, the relative D1R2 bridge frequency did not differ between substages in wild type, but was observed more frequently in zygotene- and pachytene-like Sycp1$^{-/-}$ spermatocytes compared to leptotene, although this was not significant (fold difference 2.12, p = 0.091) (Fig 3D and 3E).

Special attention was given to the occurrence of what could be considered as paired configurations, defined as a separated twin set of partially overlapping RAD51 and DMC1 nanofoci (Fig 3A: "paired" and S4 Fig). These should be mostly represented in the D2R2 subgroup. However, only 34 of the total of 142 D2R2 foci in the wild type have a "paired" appearance and the majority of these also fell in the "simple" or "complex" category, meaning that there was a major nanofocus of both RAD51 and DMC1 that were partially overlapping, and then two separate and relatively small satellites; one for RAD51 and one for DMC1 (all binary images of the wild type D2R2 foci are shown in S4 Fig). The overall frequency of paired configurations appears similar in both wild type and Sypc1$^{-/-}$ spermatocytes and never exceeded 8% of the total (Fig 3B).

Finally, a small rather constant fraction of the foci contained only separate RAD51 and DMC1 nanofoci (Fig 3A: "separate", and 3B).

The results of this manual nanofoci shape analyses confirmed the notion that paired RAD51-DMC1 co(nano)foci occurrence is rare, and confirmed the major contribution of D1R1 nanofoci. It also hinted towards biological relevance of the bridged D2R1 and D1R2 structures, since their relative frequencies showed different dynamics as prophase progressed different in Sypc1$^{-/-}$ compared to wild type spermatocytes. Given the high relative frequencies of the D1R1, D2R1 and D1R2 configurations in both wild type and Sypc1$^{-/-}$ spermatocytes, we next focussed on these three groups, leaving the more complex DxRy configurations aside based on the observations that most of these resemble one of these three groups with the addition of one or more extra "satellite" structures. We thus quantified the dynamics of nanofoci stoichiometry, and performed further analyses of nanofoci organisation within the D1R1, D2R1 and D1R2 groups using computational approaches.

## Temporal analysis of D1R1, D2R1, and D1R2 configurations during meiotic prophase

In both wild type nuclei and Sycp1$^{-/-}$ nuclei, the D1R1 configuration was the most abundant configuration at leptotene, suggesting that this is an early configuration (Fig 4A and 4B). In

the transition to zygotene in the wild type, a reduction of the D1R1 configuration fraction (p = 0.01) was observed, parallel to a 2-fold increase in the fraction of D2R1 foci, but the latter did not reach significance (Fig 4C, S2 Table, p = 0.12 (two-sided t-test)). In contrast, the D1R2 fraction remained constant. In $Sycp1^{-/-}$ spermatocytes, the average number of D1R1 foci per cell was higher in pachytene-like compared to zygotene-like nuclei (Fig 4B, S2 Table p = 0.017 (two-sided t-test)), and the fraction of D1R1 foci did not display a significant change, although it also tended to increase from zygotene to pachytene (p = 0.068 (two-sided t-test)) (Fig 4D, S2 Table). The number of D2R1 foci remained constant during the different analysed stages of the $Sycp1$ knockout (Fig 4B), but the fraction of D2R1 foci decreased as prophase progressed in the knockout (Fig 4D, p = 0.03 between pachytene and leptotene (two sided t-test, S2 Table). In contrast, the D1R2 foci number and fraction tended to increase in pachytene compared to zygotene (not significant, p = 0.06 (number) and p = 0.18 (fraction) (two-sided t-test), S2 Table). Next, we determined the fraction of foci that was on synapsed versus unsynapsed axes for each wild type zygotene nucleus and compared this to the distribution of D1R1, D2R1, and D1R2 foci in the same nuclei. Interestingly, compared to the distribution of all foci, the D1R1 configuration was slightly enriched on synapsed axes while the D1R2 configuration localized preferentially on unsynapsed axes in wild type zygotene nuclei, (p = 0.017, p = 0.037 respectively, two-sided paired t-test, Fig 4E, S2 Table).

In general, we did not observe any overt specific distribution pattern of the different configurations relative to each other along the SC at the different stages of meiotic prophase (S5A Fig).

## Asymmetrical distribution of RAD51 and DMC1 nanofoci relative to each other and to the axial/lateral elements of the SC

To investigate the spatial organization of nanofoci in the most frequently occurring configurations further (D1R1, D2R1 and D1R2), we determined the center of mass of every nanofocus in each ROI and measured the distance between the center of each RAD51 nanofocus(s) and each DMC1 nanofocus(s) (Fig 5A and 5B). Interestingly, minimum distances coherently clustered at approximately 70 nm (wild type/$Sycp1^{-/-}$;68.4±1.2 SEM/75.8±1.1 SEM) for all analysed foci configurations in wild type and $Sycp1$ knockout nuclei. Thus, almost all foci that contain more than one RAD51 and/or DMC1 nanofocus, contain at least one RAD51 and one DMC1 nanofocus in close proximity to each other, the minimal distance averaging at ~70 nm, (Fig 5A). Since only a single nanofocus is present for each of the individual recombinases in the D1R1 group, the distribution of the maximum distance was the same as for the minimum distance. Importantly, it completely overlapped with the first peak of the distribution of maximum distances of all configurations. This shows that all foci with more than one RAD51 and/or DMC1 nanofocus, always have at least one RAD51-RAD51, DMC1-DMC1, or RAD51-DMC1 nanofoci distance that is larger, with an average distance of around 300 nm (wild type/$Sycp1^{-/-}$;287.4±2.7 SEM/308.6±2.8 SEM (Fig 5B).

This observation of asymmetry allowed us to define close and far nanofoci in both the D2R1 and the D1R2 configurations. Interestingly, we observed a large close nanofocus and a small far nanofocus irrespective of whether two RAD51 or two DMC1 nanofoci were present (Fig 5C–5E). Thus, the measured larger distance between the two DMC1 or RAD51 nanofoci in the D2R1 and D1R2 configurations can be interpreted as more spatial separation, and is not caused by a second structure that is very spread out in its localisation pattern (since the far nanofocus is always relatively small). DMC1 area sizes of the close nanofoci and single nanofoci are all rather similar in wild type nuclei, and the same holds true for close and single RAD51 nanofoci. Still, the areas of these large nanofoci were decreased in zygotene in the

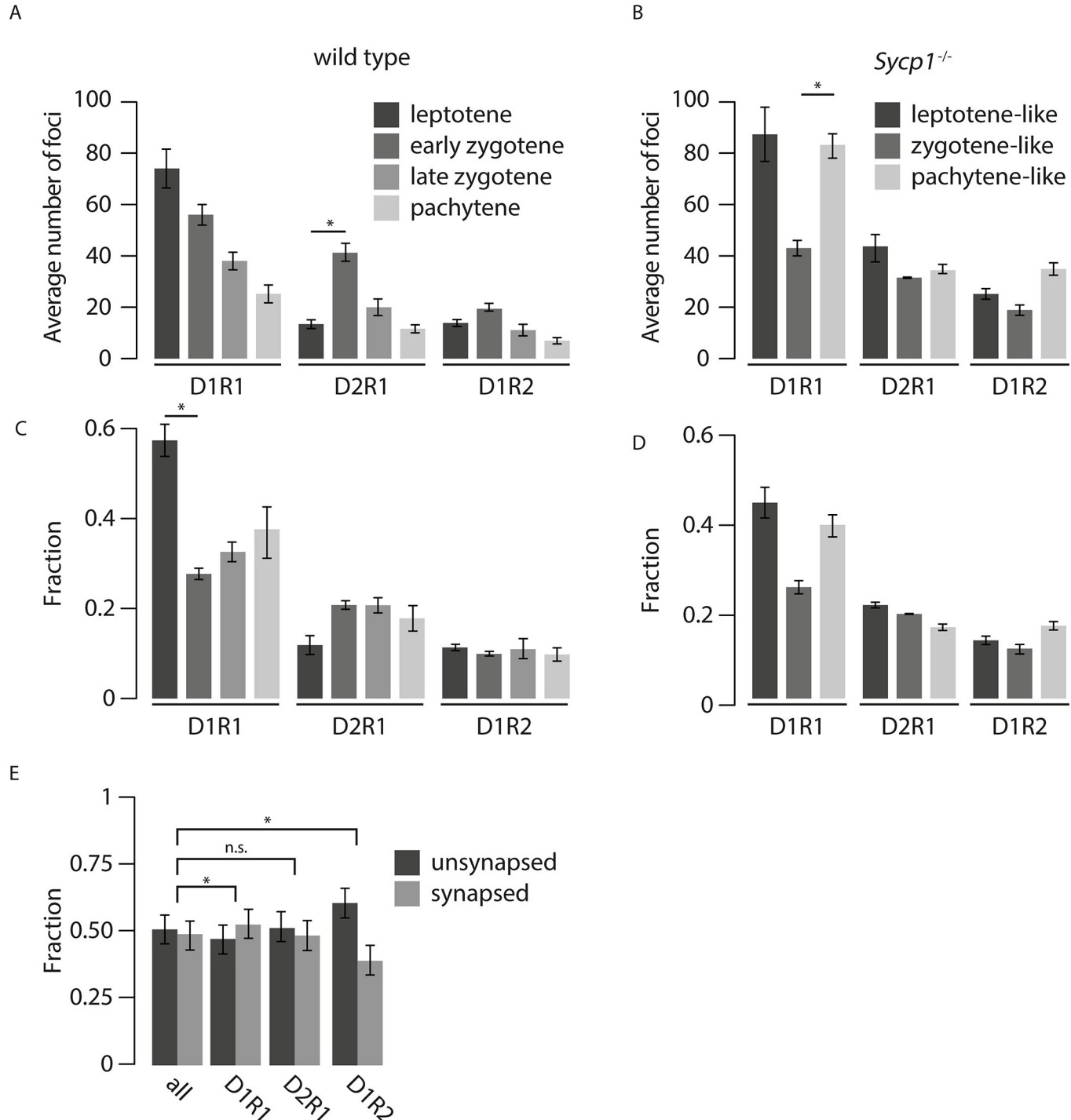

**Fig 4. Dynamics of D1R1, D2R1, and D1R2 foci numbers during progression of meiotic prophase in wild type and Sycp1⁻/⁻ spermatocytes.** Average number (A,B) and fraction (C,D) of D1R1, D2R1, and D1R2 foci per cell per stage for wild type (A, C) and Sycp1⁻/⁻ (B, D) spermatocytes. E) Fraction (right) of all foci, compared to the fraction of D1R1, D2R1 and D1R2 foci on synapsed or unsynapsed synaptonemal complexes, per nucleus at the zygotene stage. Error bars indicate SEM, p-values can be found in S2 Table.

D1R1 foci (Fig 5C, S2 Table), and the same was observed for the large DMC1 nanofocus in the D2R1 configuration. In addition, the far-DMC1 nanofocus in the D2R1 displayed a small but

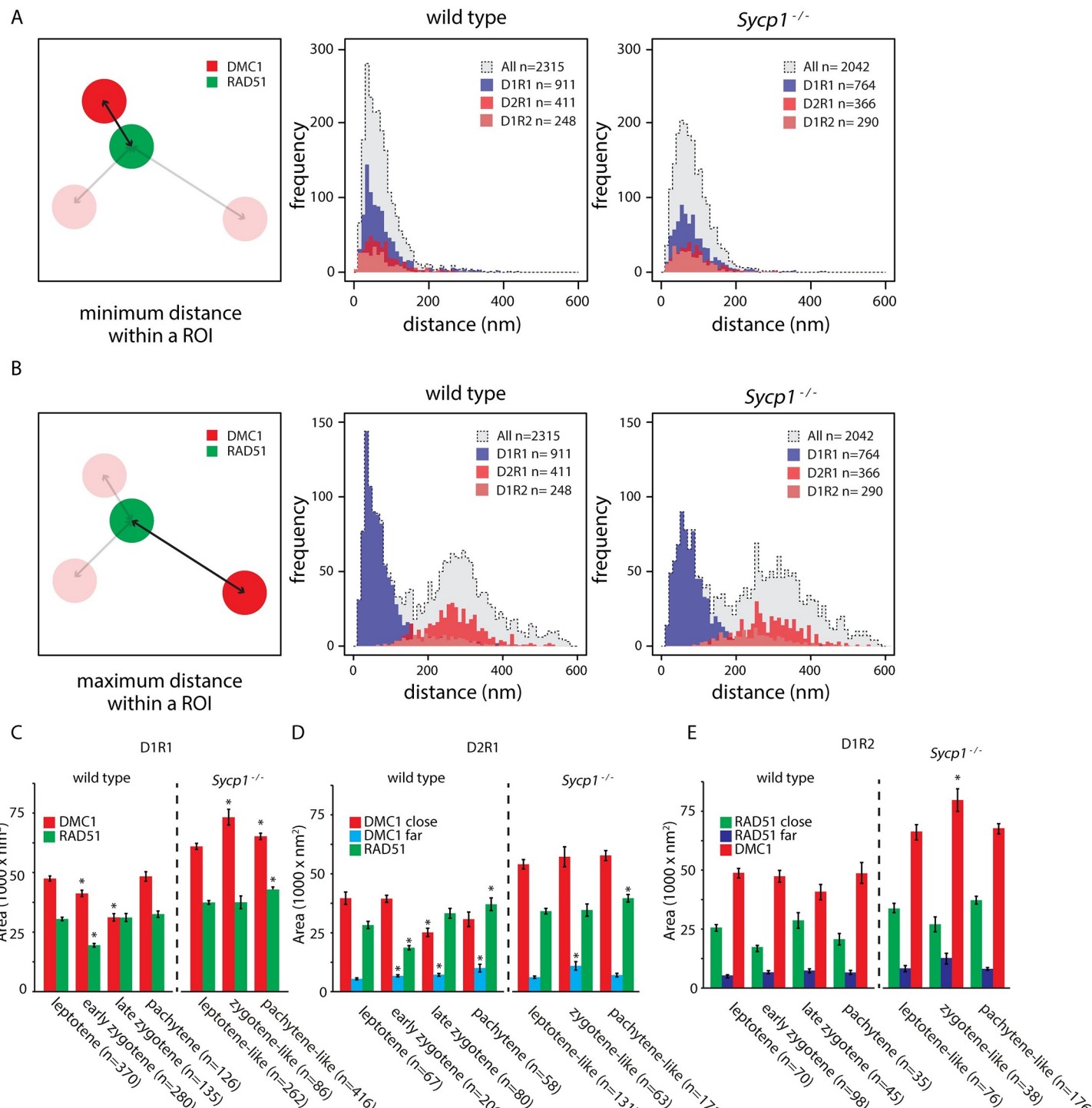

**Fig 5. Distances between DMC1 and RAD51 nanofoci, and area occupancy.** A) Distribution of the minimum distances between the center of mass of RAD51 and DMC1 nanofoci in wild type (middle panel) and Sycp1$^{-/-}$ (right panel) foci. Dashed lines with grey fill represent all foci, the D1R1, D1R2 and D21R1 subgroups are depicted in blue, light red, and red histograms, respectively. B) As in A) but maximum distances are depicted. C) Area of RAD51 and DMC1 nanofoci in D1R1 subgroup. Error bars indicate SEM, asterisks indicate significant difference compared to leptotene (p<0.05). n indicated number of foci. D) As in C) but area of RAD51 and DMC1 close and far nanofoci in D2R1 subgroup are shown. E) As in D) but area of RAD51 and DMC1 close and far nanofoci in D1R2 subgroup are shown. p-values can be found in S2 Table.

gradual increase in size as meiotic prophase progressed (Fig 5D, S2 Table). Of note, RAD51 area sizes and DMC1 area sizes did not change during prophase for the D1R2 configuration.

In *Sycp1*$^{-/-}$ spermatocytes, no consistent patterns in area size changes as prophase progressed were apparent (Fig 5C–5E).

In the wild type nuclei, we next analysed the distance of the center of mass of all RAD51 and DMC1 nanofoci to a line manually drawn through the center of the SYCP3 signal in zygotene and pachytene nuclei (Fig 6A and 6B). When all foci were analysed together, we observed that DMC1 was somewhat further away compared to RAD51 (Fig 6C, p = 0.00116 (Mann Whitney U test), S2 Table). When the D1R1, D2R1 and D1R2 foci were analysed separately, a significantly larger DMC1 distance was only observed for the far DMC1 nanofocus in the D2R1 configuration (Fig 6C, p = 0.0075 (Mann Whitney U test) S2 Table).

## Consensus patterns of the spatial organization in D1R1, D2R1 and D1R2 foci

One factor that will contribute to the observed variation in the organization of the individual images is the representation of three-dimensional structures onto a two-dimensional image. To obtain more insight in the actual structure of the three main DxRy configurations, we used alignment by rotation to be able to detect possible consensus patterns in D1R1, D2R1, and D1R2 foci (Figs 7 and 8). For the D1R1 group, the DMC1 nanofocus was used as an anchor point and the RAD51 nanofocus was used for the rotation. We rotated the structures so that the center of the RAD51 nanofocus was aligned along the vertical axis above the DMC1 nanofocus. Then we generated a single fused image of all aligned foci, pooled from the nuclei that were at a specific stage of meiotic prophase. We observed from these images that the RAD51 and DMC1 nanofocus partially overlap, but that the overlap decreases while meiosis progresses, while the measured distance between the two nanofoci increases from leptotene to zygotene (p<0.001, two-sided t-test) (Fig 7A and 7C and S2 Table). In *Sycp1*$^{-/-}$ D1R1 foci, the overlap appears also reduced at the zygotene-like stage, relative to the leptotene-like stage, but increased again at pachytene (Fig 7B). Accordingly, the RAD51-DMC1 distance increases only transiently at the zygotene-like stage (p = 0.004, two-sided t-test, Fig 7C, S2 Table). We observed no differences in distances between nanofoci within configurations on synapsed versus unsynapsed axes (S5B Fig).

For D2R1 we used the close DMC1 nanofocus as anchor, and first rotated the RAD51 nanofocus along the vertical axis. The resultant locations of the signals of the far-DMC1 nanofocus were then highly variable at leptotene, but formed a crescent moon-shaped structure around the other two nanofoci in zygotene and pachytene nuclei (Fig 7A). As meiotic prophase progresses, the far-DMC1 nanofocus was more and more localised in a smaller region above the close-DMC1 nanofocus and the RAD51 nanofocus, showing that a relatively large fraction of the D2R1 foci has a DMC1-RAD51-DMC1 type of structure. We then aligned the two DMC1 nanofoci and assessed the RAD51 location relative to the two DMC1 nanofoci by quantifying the relative number of RAD51 localisations present in four quarters (above, below, left and right) of the image, relative to the close-DMC1 nanofocus. As expected, based on the results of the rotation with the far-DMC1 nanofocus, the highest percentage of the RAD51 signal was observed between the two DMC1 nanofoci, and more signal accumulated in the upper part of that quadrant as prophase progressed (Fig 7A). In agreement with this observation, the center of mass of the RAD51 nanofocus extended away from the closest DMC1 anchor nanofocus as cells progressed from zygotene to pachytene (p = 0.0001, two-sided t-test Fig 7D, S2 Table). The mean distance between the RAD51 and the far-DMC1 nanofocus in the D2R1 decreased as prophase progressed, but increased again in pachytene (Fig 7E, S2 Table), and the same

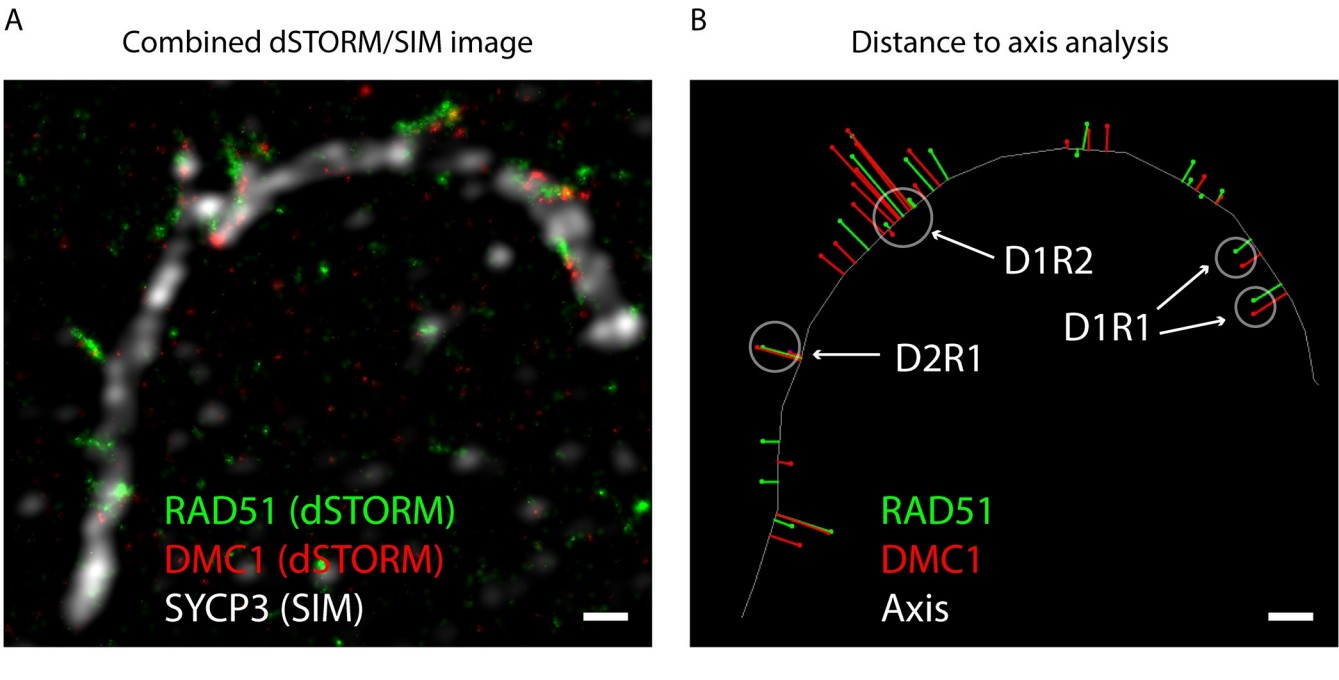

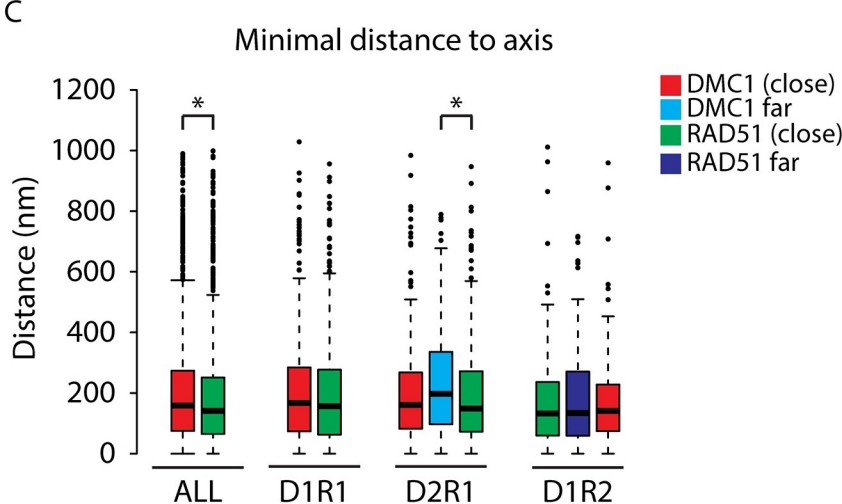

**Fig 6. Distance of RAD51 and DMC1 nanofoci to the axial and lateral elements of the SC.** A) Combined dSTORM and SIM image of a segment from a synapsed pachytene nucleus immunostained for RAD51 (green), DMC1 (red), and SYCP3 (white). B) The same area of the nucleus is shown, but here the immunosignals have been replaced by a line for SYCP3, and by dots indicating the centers of mass for RAD51 (green) and DMC1 (red). The red and green lines represent the measured distances. An example of regions where nanofoci distances were measured for D1R1, D1R2, and D2R1 foci are indicated (note that distances from another more complex ROI intersperse with the D1R2 distances). Size bars represent 500 nm. C) Box plots of minimal distances (nm) of all RAD51 and DMC1 nanofoci as well as the distances of the nanofoci of the separate D1R1, D2R1, and D1R2 classes to the axial or lateral elements in zygotene and pachytene nuclei, colour code explained in the figure. Asterisks indicate statistical significant differences, p-values are in S2 Table, the boxed region depicts the data from the first quartile to the third quartile and the whiskers extend +/- 1.5x the interquartile range (IQR = Q3-Q1; minimum value = 0). Data points lying outside this region are depicted as individual points (outliers). All DMC1 n = 1606, all RAD51 n = 2141, D1R1 n = 421, D2R1 n = 265, D1R2 n = 124.

trend was observed for the distance between the two DMC1 foci (Fig 7F). Overall, the consensus patterns in *Sycp1⁻/⁻* spermatocytes were similar, but the configurations were more variable, and few significant changes in nanofocus distances were observed (Fig 7B–7F). For example, the directionality of RAD51 towards the far DMC1 nanofocus was clear at the zygotene-like

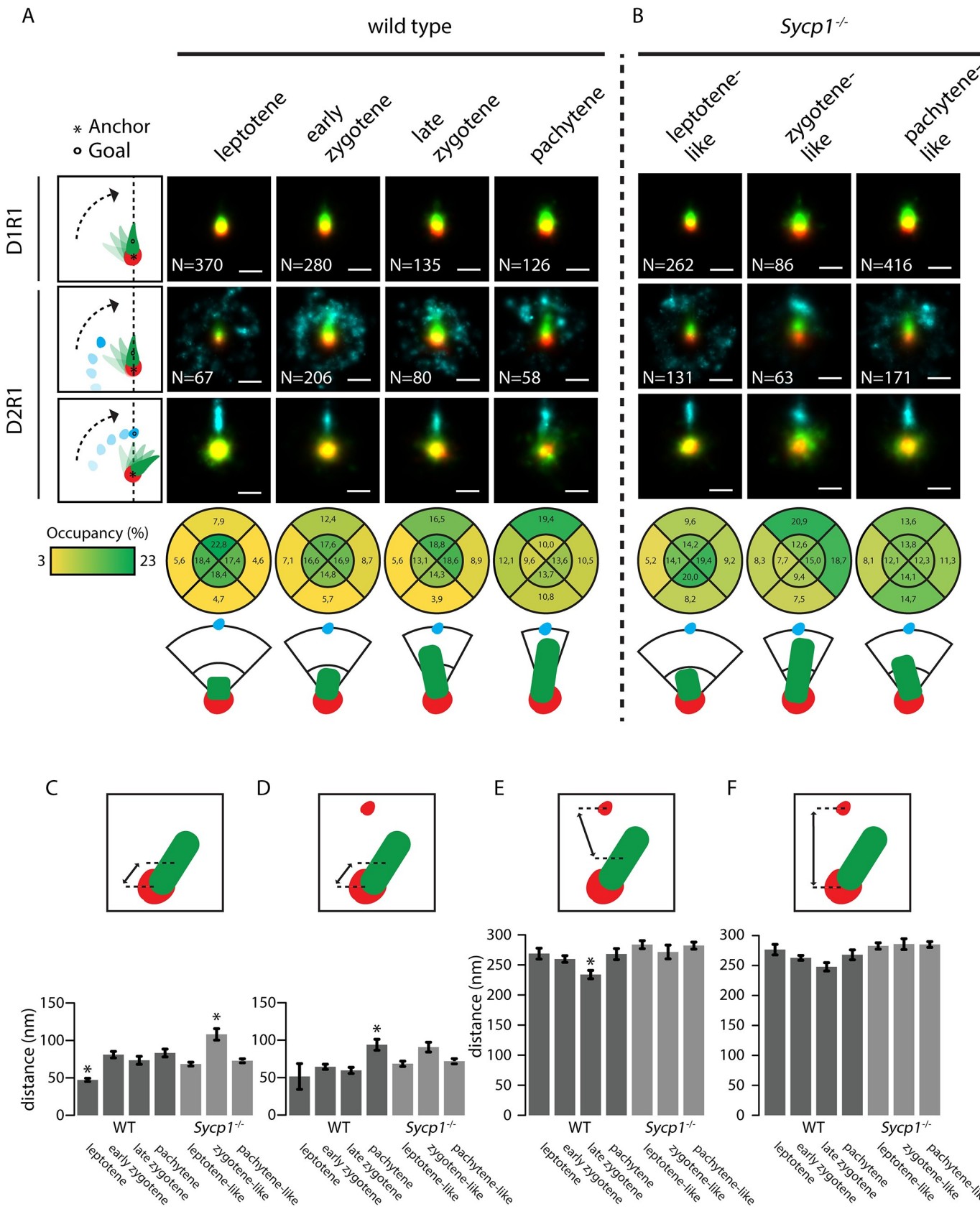

**Fig 7. Consensus patterns of D1R1 and D2R1 during meiotic prophase in wild type and Sycp1$^{-/-}$ spermatocytes.** Summed images of all rotated and aligned foci within the D1R1 and D2R1 group in wild type (A) and Sycp1$^{-/-}$ (B) per stage. Images were rotated as indicated by schematic drawings to the left of each row, whereby the anchor (*) indicates the nanofocus that is centred, and the goal (o) the nanofocus that is rotated to align along the axis. Underneath the lowest D2R1 row, the percentage of localisations for the RAD51 nanofocus in each indicated quadrant area is shown for each stage for the rotation whereby the close-DMC1 is used as anchor and the far-DMC1 as goal. A schematic interpretation of the results of the rotations is also shown. (C-F) Mean distances between the indicated nanofoci per stage in wild type and Sycp1$^{-/-}$ spermatocytes. Error bars indicate SEM. Asterisks indicate significant difference compared to all other stages (p<0.05). Scale bars represent 100 nm. p-values can be found in S2 Table.

stage, but lost at pachytene-like (Fig 7B). Furthermore, in the analyses of the distances between the nanofoci of the D2R1 configurations, the distance between the close-DMC1 nanofocus and RAD51 initially appeared to increase when cells developed from leptotene to zygotene (not significant), but in pachytene-like *Sycp1$^{-/-}$* spermatocytes, the distance was similar to what was observed at leptotene. The distance of the far-DMC1 nanofocus to RAD51 or to the close-DMC1 nanofocus was large at all stages, in contrast to the reduction observed during zygotene in the wild type (Fig 7B, 7E and 7F).

Finally, we performed the same rotation experiments for the D1R2 configuration. Interestingly, the overall organization of this configuration appeared very similar to the D2R1, including distances between nanofoci (Compare Fig 7 to Fig 8). However, in contrast to the most clear DMC1-RAD51-DMC1 organization of the D2R1 occurring in pachytene, already in early zygotene the single DMC1 nanofocus of D1R2 was most clearly located between the two RAD51 nanofoci (Fig 8A), and the DMC1 distance to the close RAD51 was already maximal at early zygotene (Fig 8C, S2 Table). No significant change in the distance to the far RAD51 nanofocus, or between the RAD51 nanofoci was observed (Fig 8C–8E). This corresponds well to the early versus late appearance of the D1R2 and D2R1 bridged structures, respectively (Fig 3B). In *Sycp1$^{-/-}$* spermatocytes, DMC1 localized more clearly in between the two RAD51 nanofoci, and this was maintained in the pachytene-like nuclei. However, the signal accumulation in the summed rotated images extended less far in the direction of the far RAD51 nanofocus compared to the wild type (Fig 8A and 8B). The increase in distance between the DMC1 and close RAD51 nanofocus was observed only transiently, in zygotene (Fig 8C, S2 Table).

## Three-dimensional simulations of the D2R1 configuration

Next, we simulated a 3D model of D2R1 configurations. In short, the model consists of three randomly generated structures reflecting the two DMC1 nanofoci and the single RAD51 nanofocus, and a random noise component. To find the best fitting model we varied the length (σ) of the RAD51 nanofocus, and the maximum angle (α) between the line connecting the two DMC1 nanofoci and the line connecting the RAD51 nanofocus to the close DMC1 nanofocus. The fixed length and variable angle reflect a possible homology searching structure as described in more detail in Materials and Methods (Fig 9A). We analysed the simulated data (discarding the z information) in the same way as the experimental data. Interestingly, around 15% of the simulated D2R1 configurations in a three-dimensional space are represented as D1R1 in the two-dimensional representations, and also a small fraction of D3R1 and D2R2 configurations were observed for simulated D2R1s. This is most likely caused by situations whereby noise detections cluster and rise just above the background, resulting in detection of an additional nanofocus. We performed rotation and alignment on the simulated D2R1 configurations in the dataset, as described above for the observed real foci. Strikingly, it can be observed that the simulated data fits best to the experimental data set if the maximum angle gradually reduces from 132° to 105° and the length of RAD51 gradually increases from 80 to 144 nm going from leptotene to pachytene (Fig 9C and 9D). Comparing the simulations to the *Sycp1$^{-/-}$* D2R1 rotations, it appears that the degree of rotation freedom for the close-

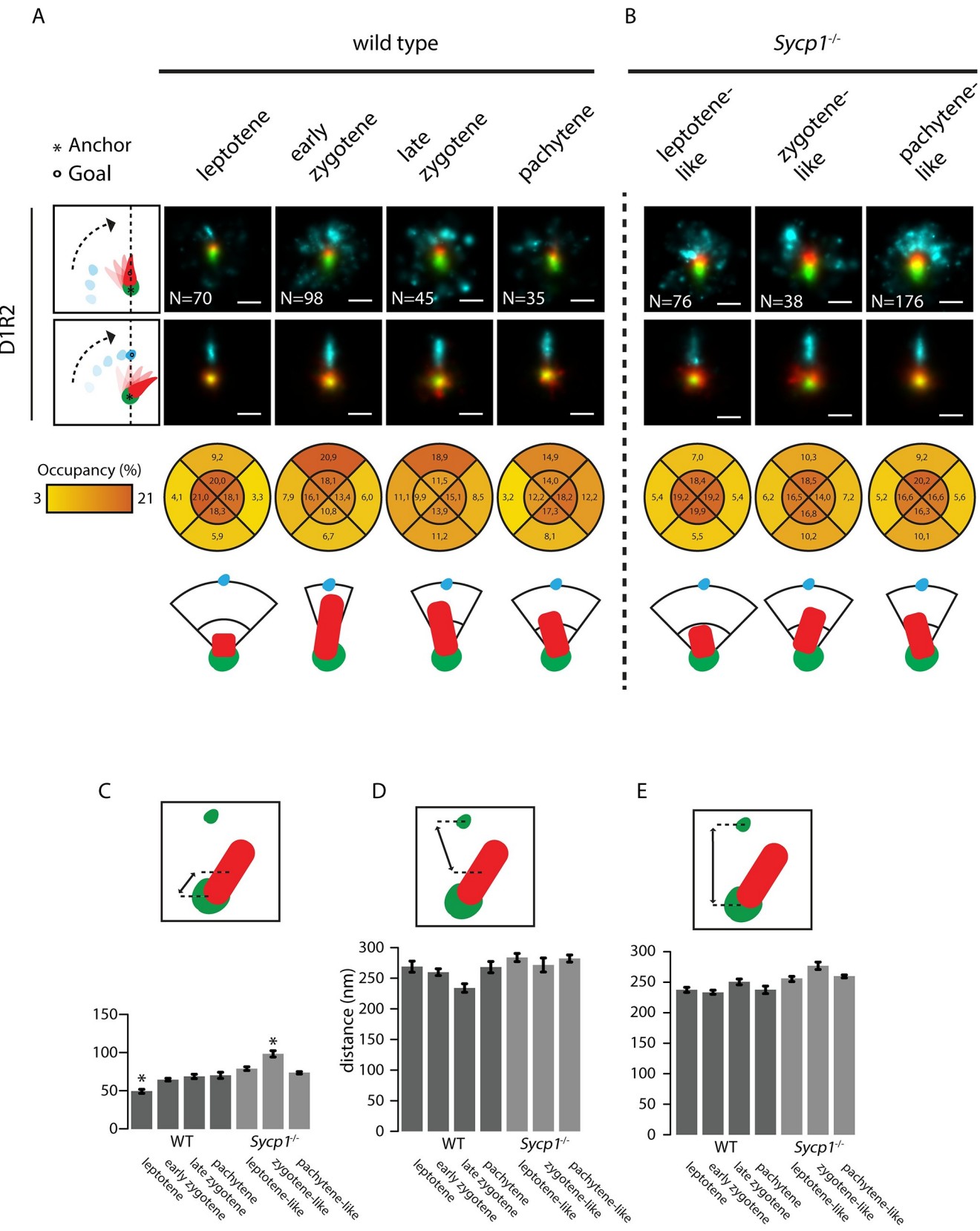

**Fig 8. Consensus patterns of D1R2 during meiotic prophase in wild type and Sycp1<sup>-/-</sup> spermatocytes.** Summed images of all rotated and aligned foci within the D1R2 group in wild type (A) and Sycp1<sup>-/-</sup> (B) per stage. Images were rotated as indicated by schematic drawings to the left of each row, whereby the anchor (*) indicates the nanofocus that is centred, and the goal (o) the nanofocus that is rotated to align along the axis. Underneath the lowest D1R2 row, the percentage of localizations for the DMC1 nanofocus in each indicated quadrant area is shown for each stage for the rotation whereby the close-RAD51 is used as anchor and the far-RAD51 as goal. A schematic interpretation of the results of the rotations is also shown. (C-E) Mean distances between the indicated nanofoci per stage in wild type and Sycp1<sup>-/-</sup> spermatocytes. Error bars indicate SEM. Asterisks indicate significant difference compared to all other stages (p<0.05). Scale bars represent 100 nm. p-values can be found in S2 Table.

DMC1-RAD51 nanofocus combination relative to the DMC1-DMC1 axis is larger than in wild type at the leptotene-like and pachytene-like stages, but actually more restricted in the zygotene-like nuclei, for which a maximal rotation angle of 105 degrees and a length of 144 nm fitted best.

## Discussion

We simultaneously determined the localisation of the recombinases RAD51 and DMC1 at nanoscale resolution in more than 4000 DSB foci in 18 wild type and 10 *Sycp1*<sup>-/-</sup> spermatocytes. We distinguished early, intermediate and late stages of meiotic prophase by co-staining of the synaptonemal complex. Together, this allowed us to reconstruct generalised RAD51 and DMC1 distribution patterns within repair foci as they progress through meiotic prophase.

RAD51 and DMC1 filaments are expected to form elongated structures, based on super-resolution images of RAD51 in somatic cells [26,27]. The maximal length of the RAD51 and DMC1 nanofoci in all observed configurations reached an average of around 140 nm in pachytene, based on our simulations, but the maximal length of the most stretched RAD51 or DMC1 nanofoci was found to be around 250–300 nm. This is comparable to the previously observed elongated RAD51 structures in fixed somatic cells using dSTORM [26]. Haas et al., [27] observed an average maximal length of RAD51 nanofoci of around 160 nm, more similar to the average simulated length we observed. It should be noted that we performed our analyses on binary images, where clusters are generated from the kernel density plots of each ROI, with a threshold at a specific localisation density. In doing so, we reduced complexity and inevitably lost some information. However, this approach allowed systematic comparison of the results between images in the same meiotic stage as well as in different stages. Although there might be some internal structure visible in the foci images shown in Fig 2F, it should be noted that the precision of each localisation event varies, and affects the gaussian distribution-based representation of the dSTORM data. In addition, the resolution is limited by the sizes of the primary and secondary antibodies, which is expected to add around 20–40 nm in X and Y direction in our 2D images [28,29]. Moreover, variability in binding may influence the actual "shape" generated by the gaussian distribution generated from the localisation events coming from the fluorophores attached to the secondary antibodies. Therefore, we feel that the binary images provide an optimal representation of the actual areas covered by the recombinases and bound antibodies. An *in vitro* filament of RAD51 with a length of 100 nm covers approximately 200 bp of ssDNA [30]. Given the current estimates of ssDNA track lengths in meiotic recombination (~500–1000 bp [31]), it seems reasonable that most of what we observe would represent actual binding of the recombinases to ssDNA, however our data also suggests that neither RAD51 or DMC1 cover the entire resected DNA in an fully extended filament. Furthermore, we certainly cannot exclude that some nanofoci represent (transient) associations with chromatin, or with dsDNA. Similarly, in yeast, dSTORM analyses of meiotic RAD51 and DMC1 foci revealed a focus length of around 115 nm for both proteins, which was proposed to cover around 100 nucleotides of single-stranded DNA, based on the known in vitro DNA binding data of the recombinases [13].

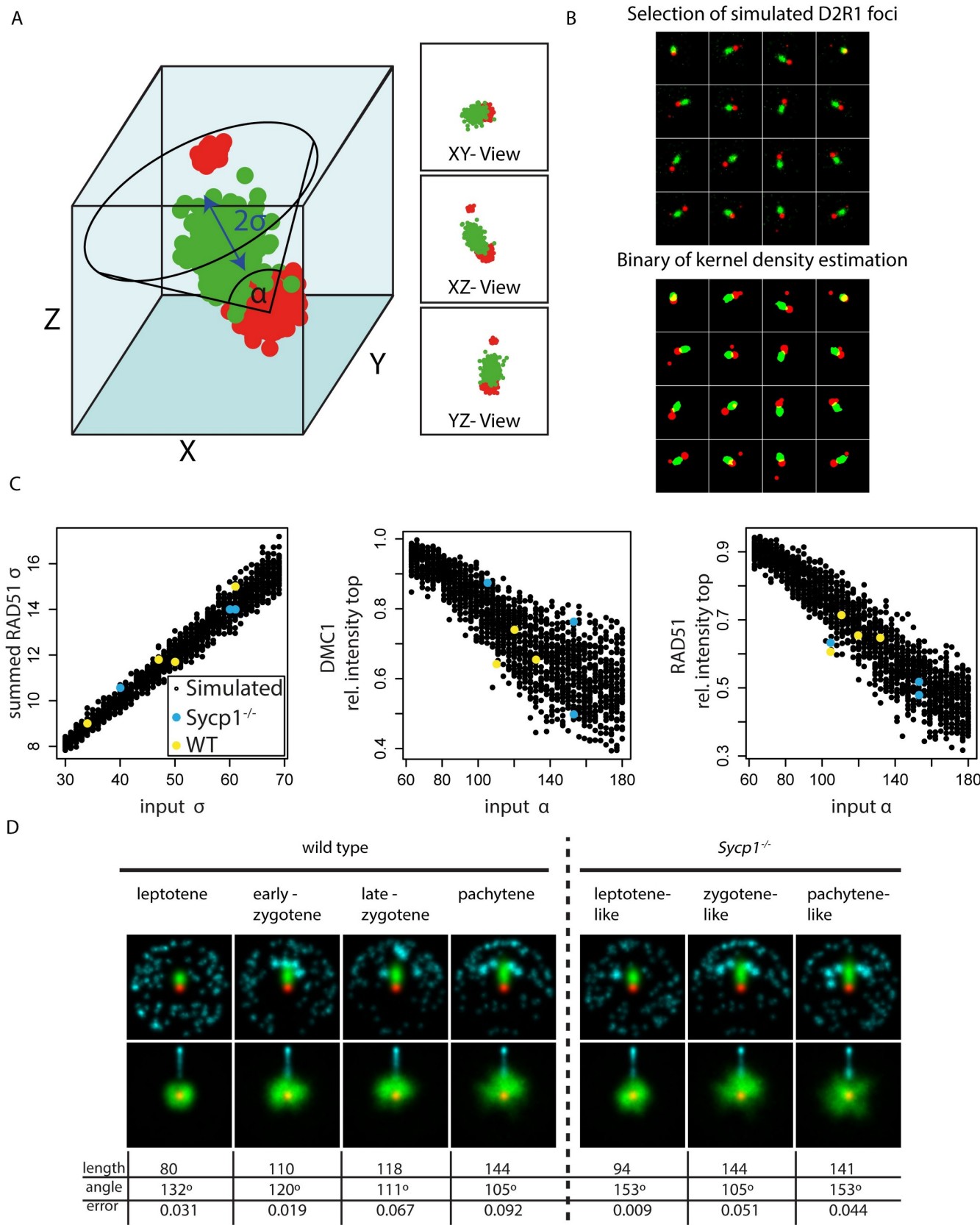

**Fig 9. Simulations of D2R1 rotations.** A) Model of D2R1 foci in three dimensions, where the alpha indicates the maximum angle relative to the DMC1-DMC1 axis, and the sigma the length of the major axis of the RAD51 nanofocus. B) Selection of simulated foci using one model randomly positioned in space and visualised in two dimensions (top), and corresponding binaries of kernel density estimation (bottom). C) Measured RAD51 length in summed 2D images (left), DMC1 intensity in the top half (middle), and RAD51 intensity in the top quadrant for all simulated foci (right), where each point represents an assembly from 200 aligned foci, and RAD51 was rotated above the center in the left and middle graph, and the far DMC1 was rotated in the right graph. Coloured points represent measured values from experimental data from both wild type (yellow) and *Sycp1*[-/-] (blue) nuclei at the stages analysed. D) Summed images (top; RAD51 rotation, bottom; far DMC1 rotation) of simulations that fit best to experimental data, length (full width half maximum: 2.355σ), angle and error are indicated.

## A close association of a large RAD51 and DMC1 nanofocus as predominant configuration in DSB repair foci

Since the D1R1 configuration was observed most frequently, and similar structures were also the major component in more complex nanofocus combinations, the D1R1-configuration represents the main form of RAD51 and DMC1 accumulation at DSB foci. The D1R1 configuration may represent asymmetric loading of each recombinase to one of the two ends of the DSB, or represent loading of both on only one end of a DSB. We hardly observed configurations that could be considered to be paired D1R1 nanofoci configurations, contrary to what might be expected based on observations in yeast [13], and from the symmetric loading of DMC1 observed in ChIP-seq data of mouse meiotic hotspots [32]. Also, in the nematode *C. elegans*, where chromosome pairing precedes meiotic DSB formation, the majority of RAD51 foci could be resolved as paired structures using structured illumination microscopy [33]. However, it might be suggested that if D1R1 configurations represent a single end of the DSB, the distance to a paired other DSB (D1R1) end would be on average 800 nm (based on the nearest neighbour analysis), but still highly variable, precluding visible paired occurrences of D1R1 structures. This could then be considered analogous to the paired co-foci observed in yeast and *C. elegans* [13,33]. Alternatively, the other DSB end could also be occupied by other ssDNA binding proteins, while a combination of these two situations may also occur.

## D2R1 and D1R2 represent DSB intermediates with asymmetric loading of RAD51 and DMC1

The similarity of the DMC1 and RAD51 nanofoci that are closest to each other in D2R1 and D1R2 to the D1R1 configurations in terms of size and proximity, and the decreasing frequency of the latter, together suggest that the D1R1 may evolve into a D2R1 or D1R2 configuration. The additional nanofocus at longer distance from the main DMC1-RAD51 entity could then result from new loading of DMC1 or RAD51, or from splitting of the respective nanofocus into two independent nanofoci that stabilizes at a distance of 200–250 nm.

The maximum area of the far RAD51/DMC1 nanofocus is more than 10-fold smaller than the areas occupied by the adjacent close DMC1 and RAD51 nanofoci. So, either the far nanofoci may be somehow compacted, or represent binding of recombinase to a shorter stretch of (ss)DNA or chromatin. It is interesting to note that in the protist Tetrahymena, it has been suggested that RAD51 filaments are extremely small, forming no visible foci, whereas DMC1 foci are observed and both proteins are required for functional processing of the meiotic DSBs [34]. Although this appears to be an example of extremely asymmetric behaviour of RAD51 and DMC1, our current observations suggest that such small filaments of either RAD51 or DMC1 may also form in other eukaryotes.

Similar to the D1R1, the large close DMC1 and RAD51 nanofoci in D2R1 and D1R2 may represent binding to the same DNA (single-stranded or double stranded) molecule, or to the different ends of the DSB. The fact that the distances of the two close nanofoci, to the far nanofocus in D1R2 and D2R1 foci are very similar in these two configurations supports the idea

that there is some form of physical coupling between the D1R1 moiety and the additional RAD51 (D1R2) or DMC1 (D2R1) nanofocus, and also that the D2R1 and D1R2 configuration represent similar chromatin/DNA conformations/repair intermediates. The "bridged" structures that were observed for both D1R2 and D2R1 also support this notion. The D2R1 bridge was observed mainly in pachytene. This structure, as well as its timing are recapitulated by the lengthening of the RAD51 nanofocus, and increased frequency of DMC1-RAD51-DMC1 alignment as prophase progresses in the rotation analyses. D1R2 bridges were found as more early structures, that preferentially locate on unsynapsed chromatin.

Interestingly, in our analyses of the distance of RAD51 and DMC1 nanofoci to the axial/lateral elements of the SC, we observed that DMC1 nanofoci were further away from the axes than the RAD51 nanofoci. In combination with the recent ChIP-seq data, that indicate that DMC1 would be loaded more towards the 3'end of the single-stranded DNA compared to RAD51 [32], and previous suggestions along these lines based on differences in biochemical properties of the two proteins [35], it might be suggested that our observations would be consistent with the hypothesis that DMC1 filaments would be held further away from the axes because of their presence at the ends of the filaments and their homology probing function. However, this hypothesis needs to be tested in more detail in future experiments that involve three-colour STORM analyses and include other components such as DNA, and/or functional tests.

## The number and organization of the RAD51 and DMC1 nanofocus combinations are affected in $Sycp1^{-/-}$ spermatocytes

Our high-resolution analyses revealed an increased number as well as relative frequency of D1R1 configurations in the pachytene-like $Sycp1^{-/-}$ nuclei compared to zygotene-like nuclei. Recent data indicate that when synapsis is not achieved, feedback mechanisms may act locally to maintain SPO11 activity in unsynapsed regions [36–38], which is in agreement with the increased frequency of early recombinase configurations in late-stage $Sycp1^{-/-}$ spermatocytes. We also observed an increased frequency of D2R1 configurations in leptotene-like nuclei, in comparison to the wild type, which can be attributed to the fact that when a true synapsed structure cannot be formed, initial alignment and pairing will be less stable, and cells that should be in zygotene will still appear as leptotene in the $Sycp1^{-/-}$ nuclei. The results of the rotation analyses and distance measurements throughout prophase in the knockout indicate that the D2R1 configuration initially appears to form and proceed as normal, but then a destabilization occurs, leading to frequencies of the D2R1 and D1R2 foci at pachytene-like stage that are more similar to those observed in wild type leptotene cells. This also fits well with a clear increase in D1R2 bridges observed in $Sycp1^{-/-}$ nuclei. It is tempting to speculate that in the absence of SYCP1, the lack of SC formation favours D1R2 structures, and that this is somehow coupled to reduced D2R1 formation/stability. In addition, the data support the previously reported longer persistence of DSB induction.

## Concluding remarks

Our super-resolution dual colour dSTORM approach allowed direct comparison of the localization of RAD51 and DMC1 relative to each other. We provide the first evidence for the presence of a major structure consisting of a single relatively large nanofocus of both RAD51 and DMC1 in close proximity to each other in the majority of mouse meiotic DSB repair foci. Additional, smaller nanofoci of either recombinase are often present, and the fact that the total number of nonoverlapping nanofoci exceeds two in ~20% of the foci indicates that either multiple non-overlapping nanofoci occupy different segments of ssDNA, as suggested by Bishop

et al., [13], or some nanofoci represent binding to dsDNA, or chromatin, or background, since maximally two DSB ends are expected to be available for binding within a single ROI. We favour the hypothesis that the D1R1 configuration mostly represents formation of two adjacent filaments of RAD51 and DMC1 on the same molecule. This then suggests that one DSB end is often not bound by the recombinases, or epitopes are hidden due to differential conformations of the two ends, or the two ends are paired but far apart (average 800 nm), with a wide variety in distances, precluding visible formation of paired co-foci.

This single-cell, and single repair focus approach revealed that there is enormous variety in the types of structures formed, in a more or less stochastic manner. We suggest that regulatory mechanisms act to stabilize or destabilize certain structures to eventually allow progression of repair using either the sister chromatid or homologous chromosome at each site, depending on local constraints. Configurations that we observe at low frequencies may still be functionally relevant, and further studies will be required to explain the observed structures in terms of actual repair intermediates. These may involve three-dimensional super-resolution imaging of repair proteins in combination with visualization of DNA. In addition, the experimental combination of meiosis-defective knockout mouse models with super-resolution microscopy provides a promising new approach to study the dynamics of mouse meiotic recombination and meiotic defects at the molecular level.

## Materials and methods

### Ethics statement

All procedures involving animals were in accordance with the European guidelines for the care and use of laboratory animals (Council Directive 86/6009/EEC).

All animal experiments were approved by the local animal experiments committee DEC Consult (protocol numbers: EMC2716, EMC3267, and EMC3201), and animals were maintained under supervision of the Animal Welfare Officer.

### Animals

Two wild type (5–10 weeks old) and two *Sycp1* knockout (12 weeks old) mice (previously described [15]) were killed using $CO2/O2$. Mice were socially housed in IVC-cages with food and water ad libitum, in 12-h light and dark cycles.

### Meiotic spread preparation and immunofluorescence

Spread nuclei for immunocytochemistry and confocal analyses were prepared as described [39]. For dSTORM and 3D-SIM analyses the same method was used, but cells were spread on 1.5 thickness high-precision coverslips (170±5 μm), previously coated with 1% poly L-lysine (Sigma). Slides were immunostained with the antibodies described below in 2 experiments to collect images for the nearest neighbour analyses. Coverslips were stained with antibodies mentioned below in six separate staining experiments for dSTORM and 3D-SIM analyses as follows:

- Four experiments to collect the images of the 18 nuclei presented in S2 Fig.

- Two experiments to collect the images of 10 *Sycp1* knockout nuclei presented in S3 Fig.

Before incubation with antibodies, slides or coverslips were washed in PBS (3x10 min), and non-specific sites were blocked with 0.5% w/v BSA and 0.5% w/v milk powder in PBS. Primary antibodies were diluted in 10% w/v BSA in PBS, and incubations were overnight at room temperature in a humid chamber. Subsequently, slides or coverslips were washed (3x10 min) in

PBS, blocked in 10% v/v normal swine serum (Sigma) in blocking buffer (supernatant of 5% w/v milk powder in PBS centrifuged at 14,000 rpm for 10 min), and incubated with secondary antibodies in 10% normal swine serum in blocking buffer overnight at room temperature. Finally, slides or coverslips were washed (3x10 min) in PBS (in the dark) and embedded in Vectashield containing DAPI (slides) or immediately used for imaging 3D-SIM and dSTORM.

### Antibodies

For primary antibodies, we used goat antibody anti-SYCP3 (R&D Systems), mouse monoclonal antibody anti-DMC1 (Abcam ab11054), and a previously generated rabbit polyclonal anti-RAD51 [40]. For secondary antibodies, we used a donkey anti-rabbit IgG Alexa 488/647, donkey anti-mouse IgG Alexa 488/647, and donkey anti-goat Alexa 555 (Molecular Probes).

### Confocal imaging

Immunostained spreads were imaged using a Zeiss Confocal Laser Scanning Microscope 700. This microscope is equipped with four lasers with wavelengths of 405 nm, 488 nm, 555 nm and 639 nm. All images were made using a 63x objective immersed in oil with a numerical aperture of 1.40 and a pinhole set at 39 μm. The digital offset was set to -2, and the laser power at 2%. The gain was adjusted for each image and channel. The images are all 1024x1024 in size, averaged 4 times.

### Nearest neighbour analysis

The confocal images were analysed to determine the distribution of RAD51 and DMC1 along the synaptonemal complexes by measuring the nearest neighbour distances. Single nuclei were manually segmented, next DMC1 and RAD51 foci were detected with the ImageJ function "Find Maxima", and a noise tolerance value of 90 (DMC1) and 100 (RAD51). Each detected "maximum" represents a single pixel with associated (x,y) coordinates, and thus each focus is thereby represented by a single x and y value in the image. We then created a mask to outline the SYCP3 signals using manual thresholding (random sampling of 40 generated masks covering a single SC revealed an average width of 477nm±24 (s.d.)), and these masks were then projected onto the image of all the maxima to remove all foci outside the selected area. These masks were also used for the projection of the pixels in the random simulations (see below). The coordinates of maxima for the remaining foci were used to calculate the distances between all foci. With these distances the nearest neighbour of each focus was determined, and the distance values were exported to Excel for further analysis. The nearest neighbour distance distributions of the observed DMC1 and RAD51 foci were compared to random distributions of foci on the SC axes, using the Kolmogorov-Smirnov (KS) test. All KS test values were generated using the R function ks.test.

### Random simulation

Simulated images were created by generating random single pixel foci (maxima) until the same number of foci were generated within the nuclear area or within the boundaries of the SYCP3 signal as the number of foci that had been counted in the nucleus, or within the mask, respectively. To achieve this, foci were generated randomly in the image, and those localising outside the nuclear or masked boundary were discarded, and remaining positions were marked. This was continued until the number of marked positions corresponded to the required number of foci. This created an image with single pixel foci. To correct for the diffraction limited signal of a confocal microscope, the random image was blurred with a Gaussian

filter with a sigma value of 0.11 μm. This sigma value is approximately the standard deviation of the confocal microscope (FWHM = $2\sqrt{2\ln2}\,\sigma \approx 2.355\,\sigma$ (Weisstein, 2002)). Simulated shot noise was added by adding a value of 5 to the entire image, and subsequently adding a random value between +/- the square root of the intensity of each individual pixel. This image was then processed in the same way as the confocal images. 50 random simulations were performed for each nucleus.

## 3D-SIM and dSTORM imaging

Coverslips immunostained as described above were mounted in an Attofluor Cell Chamber (Life Technologies). For drift correction and channel alignment 100 nm Gold nanoparticles (Sigma) were added to the sample. To perform dSTORM imaging, an imaging buffer was prepared containing 40mM MEA (Sigma), 0.5mg/ml Glucose Oxidase (Sigma), 40 μg/ml Catalase (Sigma) and 10% w/v Glucose in PBS pH 7.4. Samples were incubated in the imaging buffer during the entire imaging session.

Imaging was performed using a Zeiss Elyra PS1 system. Both 3D-SIM and dSTORM data were acquired using a 100x 1.49NA objective. 488, 561, 642 100mW diode lasers were used to excite the fluorophores together with respectively a BP 495–575 + LP 750, BP 570–650 + LP 750 or LP 655 excitation filter. For 3D-SIM imaging a grating was present in the light path. The grating was modulated in 5 phases and 5 rotations, and multiple z-slices were recorded on an Andor iXon DU 885, 1002x1004 pixel EMCCD camera. dSTORM imaging was done using near-TIRF settings while the images were recorded on Andor iXon DU 897, 512x512 pixel EMCCD camera. At least 10 000 images were acquired at an interval of 33ms for Alexa 647. For Alexa 488 an interval of 50ms was used to compensate for the lower photon yield of the Alexa 488 dye. We used Alexa 488 and Alexa 647 dyes coupled to secondary antibodies to detect respectively RAD51 and DMC1 or vice versa. Using either fluorophore combination, we consistently detected ~1.5 times more localisation events for RAD51 than DMC1. As expected, we observed more localisations for Alexa 647 compared to Alexa 488, due to the more suitable photochemical properties for dSTORM of the former [41]. We chose the more efficient Alexa 647 dye to detect DMC1, that is either less abundant or less well recognized by the primary antibody compared to RAD51, and the Alexa 488 dye to detect RAD51.

## 3D-SIM and dSTORM image analysis

3D- SIM images were analysed using the algorithm in the ZEN2011 (Carl Zeiss, Jena) software. For dSTORM, individual fluorescent events were localised in the subsequent frames using a 2D Gauss fitting algorithm in the ZEN2011 (Carl Zeiss, Jena) software. Detections in subsequent frames originating from the same fluorophore were grouped. Drift was corrected using 100nm gold nanoparticles (Sigma). The same fiducials were used to align the two colour dSTORM images using an affine alignment. Dual colour dSTORM and triple colour SIM images were aligned, based on the dSTORM and 3D-SIM Alexa 647 images, using a channel alignment algorithm in the ZEN2011 software. All observed foci were manually selected based on the SIM images, and circular regions (radius of 300 nm) around the foci were selected using ImageJ within the Fiji platform [42]. For each stage and each genotype, 2–5 nuclei were analysed. Each nucleus can be viewed as a biological replicate when differences between stages are considered, whereas each focus can be considered a biological replicate when the overall properties of the foci are analysed. The single molecule localisations of the individual foci were subsequently imported into R using the RStudio GUI for further analysis [Pau, Oles, Smith, Sklyar and Huber, EBImage: Image processing toolbox for R. v. 2.13 (2013) https://bioconductor.org/packages/release/bioc/html/EBImage.html; R Development Core Team, R:

A language and environment for statistical computing. R Foundation for Statistical Computing, R Foundation for Statistical Computing, Vienna, Austria, ISBN 3-900051-07-0, http://www.R-project.org].

Selected foci that were spatially overlapping were excluded if the percentage of overlapping localisations was larger than 25% [21]. Also foci containing less than 50 localisations were excluded from further analysis.

### Foci analysis

Single molecule localisation data was used to fit a 2D Kernel Density Estimation (KDE) function [Wand, 2013, KernSmooth: Functions for kernel smoothing for Wand & Jones 2.23–10, http://CRAN.R-project.org/package=KernSmooth]. The KDE function estimates the density of localisations at a certain position in the image. The bandwidth of the density estimation was set to the approximate average localisation precision of our data: 20 nm. The 2D KDE gives a normalized density over the image. Because we are interested to determine the absolute density of localisations, the normalized density is multiplied by the number of localisations in the ROI. After fitting a 2D KDE to the data we are able to define objects by applying a threshold. The threshold was set at 5 localisation/pixel, equal to 0.2 localisations/nm$^2$. Very small nanofoci with an area covering less than 50 pixels were considered background. The resulting binary images were used to determine shape features (center of mass i.e.) [Pau, Oles, Smith, Sklyar and Huber, EBImage: Image processing toolbox for R. v. 2.13 (2013) http://bioconductor.org/packages/release/bioc/html/EBImage.html]. Manual classification as described in Fig 3 was done on all foci with at least one DMC1 and one RAD51 focus, subsequently foci were sorted in different categories (S1 Table).

Differences between foci configurations at different cell stages (Figs 3 and 4) were tested using two sample Student t-test. To test differences in the prevalence of all, versus D1R1, D2R1, and D1R2 configurations on synapsed versus unsynapsed axis (Fig 4E) a paired t-test was used. A paired test was used since the overall fraction of foci on synapsed axis will depend on the meiotic progression of the individual cells (S2 Table). Pairwise comparison between the mean values of image features from individual nuclei and separate meiotic stages was performed using an independent two sample Student t-test. A p-value below 0.05 was considered a significant difference between the two samples. The minimal distance from the axial/lateral elements was determined for each nano-focus in wild type zygotene and pachytene nuclei. A line was manually drawn through the middle of the SYCP3 signal in the SIM images and the shortest distance between the axis and the nano-focus was determined. Nanofoci with a distance >1μm were considered not near an axis. Standard box-plots were generated in R. Pairwise comparison between the mean values of DMC1, RAD51 and the specific far nanofoci was performed using a Mann Withney U test. A p-value below 0.05 was considered a significant difference. For alignment by rotation the center of mass was used to center images on the close DMC1 nanofocus for alignment by rotation. The subsequent localisations were all rotated so that either the far DMC1 or RAD51 center aligned above the (close DMC1) center. All localisations from indicated stages were pooled and rendered as an image using SMoLR [21].

### Simulation

We generated a 3D model of a D2R1 focus consisting of three distinct Gaussian distributions of 3D coordinates. The two DMC1 nanofoci are represented as globular distributions where the standard deviation (σ) of the Gaussian distribution is equal in x,y and z. RAD51 is represented as an ellipsoid distribution in which the σ of the Gaussian distribution is larger in one dimension. We used the mean number of localisations measured per nanofocus: 267, 564 and

 

51 coordinates for RAD51, close DMC1 and far DMC1 respectively. We included 50 randomly distributed background coordinates in the model. The model was organized in such a way that the 'close' DMC1 nanofocus and the RAD51 nanofocus are physically connected. The far DMC1 nanofocus was placed randomly at distance of 400 nm from the close DMC1 and the RAD51 nanofocus localises at a random angle relative to the DMC1-DMC1 axis in a three-dimensional space. We then varied the length of the main axis of the RAD51 nanofocus ($\sigma$) and the maximal angle ($\alpha$) at which the 'close' DMC1-RAD51 nanofocus combination could be positioned relative to the DMC1-DMC1 axis, and generated datasets of 200 configurations for every combination of $\sigma$ and $\alpha$. We fitted the experimental data to the simulations using 3 parameters: the $\sigma$ of a Gaussian fitted over the RAD51 signal ($\sigma$-RAD51), the percentage of DMC1 signal in the top half of the center ($\alpha$-DMC1) in the rotation where RAD51 is aligned to the top, and the percentage of RAD51 in the top quadrant ($\alpha$-RAD51) in the rotations where the far DMC1 is aligned to the top. These 3 parameters where measured in both the simulated data and the experimental data (Fig 7B). Using a least mean squares method the simulation which fits the experimental data best was determined.

## Supporting information

**S1 Fig.** Foci numbers in confocal images used for nearest neighbour distance measurement and statistical analyses (A) Foci numbers were determined automatically using FIJI as described in materials and methods. Numbers counted in each individual nucleus are shown. Horizontal bar depicts the average and error bars indicate standard deviation. B), C) Relative frequency distribution of nearest neighbour distances between DMC1 and RAD51 (left) DMC1 and DMC1 (middle) and RAD51 and RAD51 (right) in leptotene (B, n = 7 nuclei; 1003 DMC1 foci, 1157 RAD51 foci) and zygotene (C, n = 6 nuclei; 645 DMC1 foci, 590 RAD51 foci) wild type nuclei (all foci were analysed). Distances were binned in 100 nm bins, distances larger than 3.4 μm were labelled as rest. Grey bars, experimental data; red bars, simulated data (see Materials and Methods). p-values of Kolmogorov-Smirnov tests are indicated in the graphs.
(TIF)

**S2 Fig.** Analysed wild type nuclei (A) 3D-SIM images of the wild type nuclei analysed per stage. Nuclei were immunostained for RAD51 (green), DMC1 (red), and SYCP3 (white). In cases where two nuclei were imaged in the same field of view they are separated by a dashed line. Scale bars represent 5 μm. Asterisk indicates late zygotene nucleus of which foci are shown in Fig 2F (B) Bar graph showing the average number of foci from wild type spermatocyte nuclei that were analysed in dSTORM per stage (leptotene, early/late zygotene, pachytene). The number of analysed nuclei per stage is indicated to the left of each bar. Error bars indicate SEM, asterisk indicate significant difference to all other stages (p<0.05). (C) p-values for foci number comparisons between stages (yellow background; p<0.05, green background p<0.005).
(TIF)

**S3 Fig.** Analysed *Sycp1*<sup>-/-</sup> nuclei (A) 3D-SIM image of microspread pachytene-like meiotic nucleus from *Sycp1*<sup>-/-</sup> mouse immunostained with primary antibodies for RAD51, DMC1, and SYCP3, and appropriate secondary antibodies labelled with Alexa 488 (green), Alexa 647 (red), and Alexa 555 (white), respectively. The boxed region is shown to the right and the arrowheads mark regions shown below. (B) Bar graph showing the average number of foci from wild type spermatocyte nuclei that were analysed in dSTORM per stage (leptotene-like, zygotene-like, pachytene-like). The number of analysed nuclei per stage is indicated underneath each bar.

Error bars indicate SEM values. (C) p-values for foci number comparisons between stages (yellow background; $p < 0.05$). (D) 3D-SIM images of the *Sycp1*<sup></sup> nuclei analysed per stage. Nuclei were immunostained for RAD51 (green), DMC1 (red), and SYCP3 (white). (E) A compilation of all ROIs of the left zygotene-like nucleus shown in D (indicated with *), ROIs are sorted by their DxRy configuration, from most frequent to rare configuration. The images are reconstructed with plotted Gaussian distributions proportional to the precision of the individual localisations.
(TIF)

**S4 Fig. Morphological classification of all wild type D2R2 foci. All D2R2 foci are shown, classified as described in the main text.**
(TIF)

**S5 Fig. Distribution of different DxRy configurations along the chromosomes of wild type spermatocytes, and analyses of distances between DMC1 and RAD51 nanofoci on synapsed and unsynapsed axes.** (A) The ROIs defined for a wild type leptotene, early zygotene, late zygotene and pachytene nucleus immunostained for RAD51, DMC1 and SYCP3 are superimposed on the SYCP3 SIM image (white). Red ROIs correspond to D1R1, green ROIs correspond to D2R1, blue ROIs to D1R2, yellow ROIs to D2R2 and magenta ROIs to the rest group of configurations. Scale bars indicate 5 µm. (B) Mean distances between the DMC1 and RAD51 nanofoci in D1R1 and D2R1 configurations per stage in wild type spermatocytes, distributed over synapsed or unsynapsed axes. Error bars indicate SEM.
(TIF)

**S1 Table. This Excel file contains the data for each focus that was analysed in wild type and *Sycp1*<sup></sup> nuclei, as explained in Materials and Methods.**
(XLSX)

**S2 Table. This Excel file contains data used for statistical analyses relating to Fig 1, S1 Fig, Fig 2, Fig 4, Fig 5, Fig 6, Fig 7 and Fig 8.**
(XLSX)

# Acknowledgments

We would like to thank Prof. Dr. J. Anton Grootegoed (Erasmus MC, Rotterdam) for his valuable comments and suggestions to the initial manuscript draft and Sabrah Niesten (Erasmus MC, Rotterdam) for her contribution to the code components of the nearest neighbour analyses.

# Author Contributions

**Conceptualization:** Johan A. Slotman, Maarten W. Paul, Fabrizia Carofiglio, Adriaan B. Houtsmuller, Willy M. Baarends.

**Formal analysis:** Johan A. Slotman, Maarten W. Paul, Fabrizia Carofiglio, Tessa Vergroesen, Lieke Koornneef.

**Funding acquisition:** Adriaan B. Houtsmuller, Willy M. Baarends.

**Investigation:** Johan A. Slotman, Maarten W. Paul, Fabrizia Carofiglio, Tessa Vergroesen, Lieke Koornneef.

**Methodology:** Johan A. Slotman, Maarten W. Paul, Fabrizia Carofiglio, H. Martijn de Gruiter, Tessa Vergroesen, Lieke Koornneef, Wiggert A. van Cappellen, Adriaan B. Houtsmuller.

**Software:** Johan A. Slotman, Maarten W. Paul, H. Martijn de Gruiter.

**Supervision:** Adriaan B. Houtsmuller, Willy M. Baarends.

**Visualization:** Johan A. Slotman, Maarten W. Paul, Fabrizia Carofiglio, Willy M. Baarends.

**Writing – original draft:** Maarten W. Paul, Fabrizia Carofiglio, Willy M. Baarends.

**Writing – review & editing:** Johan A. Slotman, Maarten W. Paul, Fabrizia Carofiglio, H. Martijn de Gruiter, Tessa Vergroesen, Lieke Koornneef, Wiggert A. van Cappellen, Adriaan B. Houtsmuller, Willy M. Baarends.

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
