## [Decision Letter · Decision Letter 0]

7 Feb 2020

Dear Dr Baarends,

Thank you very much for submitting your Research Article entitled 'Super-resolution imaging of RAD51 and DMC1 in DNA repair foci reveals dynamic distribution patterns in meiotic prophase' to PLOS Genetics.

The manuscript was fully evaluated at the editorial level and by one external peer reviewer. We attempted to secure additional external reviews but were not successful, and have now decided to proceed on the basis of the existing external review, accompanied by additional review comments from one of us (DKP). As you will see from the comments below, we are pleased to say that the manuscript should be acceptable for publication, in principle, following revisions that address concerns noted below, but should not require further experimentation. Many of the comments from the external reviewer (with which we agree) were also noted by the editors, but are not duplicated in the editor's review comments below. 

We therefore ask you to modify the manuscript according to the review recommendations before we can consider your manuscript for acceptance. Your revisions should address the specific points made by each reviewer.

[LINK]

Yours sincerely,

Douglas K. Bishop

Guest Editor

PLOS Genetics

Gregory Barsh

Editor-in-Chief

PLOS Genetics

Comments from the Guest Associate Editor:

This is an interesting paper describing the structures formed by two recA family members, RAD51 and DMC1, during meiotic prophase in mouse spermatocytes. Unlike observations in yeast and worms, the results seem to indicate that only a single major staining focus of RAD51, and a single focus of DMC1, forms at each site of recombination rather than a pair of RAD51-DMC1 co-foci. This suggests that rather than one complex forming on each end of a DSB, there is either asymmetric loading or some other process that accounts for the observed difference. In addition to a single large staining focus for RAD51 and for DMC1, there are sometimes additional small foci of either DMC1 or RAD51 associated with the large foci. The paper also examined a mutant defective in forming the central region of the SC and identifies some minor effects of this mutant on the classes of DMC1-RAD51 structures observed. The work is meticulously done and the data is thoughtfully interpreted. Although the new insight from the work is somewhat limited, the observations will no doubt be of interest to readers of *PLoS Genetics* who study meiosis and recombination.  I am therefore in favor of accepting the manuscript for publication provided a number of concerns about the manuscript are addressed. These are listed below.

The use of the term “cluster” in place of foci in many, but not all, parts of the paper is confusing. Although I appreciate that the terms focus (singular) and foci (plural) are most frequently used to describe structures that are below the resolution limit of the microscopy method, that is not strictly the case. Given that the field is familiar with the terms ‘focus” and ”foci”, I think it best to use those terms throughout the paper. The term “cluster” is most frequently used to describe a grouping of discrete objects and using it to describe a single contiguous region of staining, albeit composed of a cluster of epitopes, will likely be as confusing to other readers as it was to me (at first). This is particularly true because much of the paper describes clusters of foci (clusters of clusters). One can describe a given focus as being small, big, elongated etc. I would entertain a proposal to use some other term that does not have the problem that the term “cluster” has, but I think using the term “focus” is most straightforward given that it is so prominent in the literature. (Note that many papers incorrectly use the term “foci” as both singular and plural. Again, the singular is “focus.”)

I was disappointed and somewhat confused by the use of a mask to eliminate foci that are not colocalized to the axial/lateral elements. This seems to me to have been a poor choice given that there are models for the homology search that invoke a searching arm that is released from the axes. One model invokes a “tentacle” long enough to search the entire volume of the nucleus and our lab’s preferred model suggests a shorter tether to the axis of up to 400 nm in yeast. I imagine that examination of SIM images did not provide evidence of the focus pairs seen in other organisms with one member of each pair off the axis. If no real care was taken to discount this possibility, that is a problem. If repeating the nearest-neighbor analysis without the mask does not reveal more pairing of foci than was the case with the mask, then the mask is not a problem. HOWEVER, this analysis of the available images needs to be done and the results reported to assure the reader that important data was not lost by the method (perhaps in a supplemental figure).  I also think the paper needs a better rationale as to why the mask was used, given that not all foci fall on the SC and there is no obvious reason to think that foci that do not colocalize with the SC are irrelevant to meiotic recombination.

Although the scope of the paper is ok as is, the paper would be of MUCH more impact and interest if it included analysis of the average minimum distance of RAD51 and DMC1 foci to the midline of Scyp3 axes. I appreciate the near impossibility of 3-color STORM, but is it not possible to use SIM images to ask if RAD51 is closer or farther from the axis than DMC1? This question is of particular interest given the work from the Donnelly group showing that DMC1 loads 3’ or RAD51 at recombination hotspots. Is DMC1, which is likely to be the active recombinase based on results from yeast and plants, held away from the axis in a manner that may facilitate pairing? It seems to me that this is an important question that you might have the data in hand to address.

Line 112. Capitalize the abbreviation for Saccharomyces.

Line 163. Why was a diameter of 600 nm chosen to define a ROI?

Line 166 and in many other places in the manuscript. Claim is made that: “Foci with multiple RAD51 or DMC1 clusters were also present, and were somewhat more frequent for DMC1 compared to RAD51.” In such instances, if would be helpful if the authors indicated the fold difference between the two frequencies and give the P value for significance of the difference. I also ask “was a similar difference seen in a least two independent repeats… in two independent experiments?  One has to be particularly concerned about this given that some of the “satellite” foci are likely to be near the detection limit, such that staining efficiency could impact the results. There are a number of similar instance of claims of small differences in the paper, as mentioned below. Are have all of these observations been repeated?

Line 187 see comment re 166.

Line 193 see comment re 166.

Line 193. “Domain” is a potentially confusing synonym. “Focus” would be more clear.

Line 205. Are the structures referred to as bridges always elongated or is the definition simply based on the relative position of the RAD51 focus in between two DMC1 foci (or vice versa)?

Line 211. It is not clear to me what distinction is being made from paired verses unpaired D2R2 foci. Aren’t D2R2 foci paired by definition?

Line 246. It is confusing that this paragraph switches from description of “minimum distance” to average distance and also that 70 nm is the typical distance of each focus center to the center of mass AND the typical distance between centers of RAD51 and DMC1 foci. This could be more clearly explained.  

Line 262 It is stated that “the measured larger distance between the two DMC1 or RAD51 clusters in the D2R1 and D1R2 configurations can be interpreted as more spatial separation.” Is there any way two things that are farther apart would not be interpreted to have more special separation? Am I missing something?

Line 265. See comment re 166.

Line 284, 288. See comment re line 166.

Line 340 How can a simulated structure be spurious? Does this mean that the R focus is arranged in such a way that it divides a single D focus in two? -ie not “detecting an additional cluster” but rather the overlap of a single focus by a narrower 51 focus makes it look like 51 is separating two distinct DMC1 foci?

Line 343. I have not seen the phrase “degree of freedom for the angle” used before, which could reflect my ignorance!  Is this alternative statement accurate? “the angle of rotation that provided the best fit to the experimental data set was…” If not, a better explanation of the manipulation used to give the best fit is needed for those, like me, with a limited math background.

Line 374. Here I think it is worth comparing the results described here to those we described for the equivalent experiment in budding yeast, where foci in yeast are a bit shorter. It is also worth mentioning that we too concluded that recombination filaments are likely shorter than ssDNA tracts.

Line 534. The term maxima may confuse people not familiar with quantitative microscopy. If there is always one maxima per focus you could just say something like  “the number of foci detected in a given image determined the number of simulated foci randomly projected in 3 dimensions to create a simulated image.” I also note that this section has a lot of technical language, but lacks specific explanation for why certain procedures were necessary—i.e. what they contributed to the validity/utility of the simulated images. I take it that the manipulations were intended to mimic characteristics of the imaging system, but I think it would be helpful to state that directly.

Line 566. It is a relatively straightforward experiment to distinguish between the two possibilities listed, i.e a quantitative western blot of titrated amount of protein.  One simply needs a little pure RAD51 and pure DMC1. Many labs share these reagents, including ours. This would allow the authors to estimate the relative amounts of each protein in the structures they detect. I will not require that this be done for publication, but it easily could be done and such information would add substantially to the paper.

I agree with the comments of the other reviewer regarding data presentation and will not repeat those comments here.

Reviewer's Responses to Questions

**Comments to the Authors:**

Reviewer #1: In this work, the authors use a combination of confocal microscopy, SIM and STORM to investigate how the recombinases DMC1 and RAD51 are organized relative to each other at recombination sites in mouse spermatocytes, with the aim of providing insight into the process of meiotic recombination. A great deal of effort has been applied to analyze the data sets using several different approaches. In addition, they also use modeling as a means to cross-validate their approach for generating consensus patterns of recombination-site DMC1/RAD51 co-clusters.

One major finding of this work is that in contrast to observations of DMC1/RAD51 co-foci in yeast meiosis and RAD51 foci in C. elegans meiosis, which tend to occur in pairs that represent both ends of a DSB, DMC1/RAD51 co-foci in mouse spermatocytes usually do NOT occur as pairs. This implies that the two ends at a given DSB site are either treated very differently in regard to timing, processing and/or protein associations, or they are typically separated by distances >500 nm. Other observations suggest that recombinase-bound intermediates may become elongated during meiotic progression, and that meiotic progression also involves progression from early intermediates that typically have a single large cohort of DMC1 adjacent to a single large cohort of RAD51 to later intermediates with one or more additional small cohorts of recombinase proteins.

It is clear that the authors have carried out a careful and thorough analysis of this valuable data set. While the amount of new insight generated regarding meiotic recombination mechanisms is relatively modest, their findings are directly relevant to the process of chromosome inheritance during meiosis, a topic that is highly appropriate for PLoS Genetics. And to their credit, the authors have not tried to oversell, which is quite refreshing in the current era. Thus, I support publication in PLoS Genetics of a revised manuscript that addresses the issues below. All of these can be addressed by revision of text, modification of figures, or reanalysis of existing data, so no new experiments would be required.

General points:

Improved resolution of figure panels:

The resolution of many figure panels in the pdf provided was of poor quality. This was true both for images and for graphs.

Frequency vs number: The term “frequency” appears to be used in multiple different ways in different parts of the manuscript, leading to some confusion, especially in cases where the numbers of foci present change between stages. Examples where this is a problem will be discussed below, but care should be taken throughout the manuscript to be precise in use of language to avoid such confusion.

Basis for staging of nuclei for Sycp1 mutant: Whereas staging of nuclei from the WT is straightforward based on the state of synapsis, this becomes challenging in a mutant in which synapsis does not occur. In order to make strong statements regarding progression and timing in the context of the Sycp1 mutant, confidence in staging is crucial. Please indicate how stages were assigned for the mutant, and the degree of confidence in these assignments.

Specific points:

Abstracts/intro:

L 51 clarify wording: repair sites often contain adjacent or overlapping single large clusters of the two proteins…

L54 synapsis (instead of pairing)

L79 suggesting (instead of indicating)

L101 use of the D1R1 terminology that is not yet introduced

Figure 1 and related text:

Please clarify how the random distributions were simulated. From the Methods, it is clear how the simulated images were treated, processed and analyzed, but it was not clear how the initial random distributions were actually generated in the first place. E.g. was an existing ImageJ plug-in used, or was there a custom script used to do this? This information will be helpful for other researchers in the field that want to do similar analyses.

For the Figure, please increase font sizes wherever possible, including making it easier to see “leptotene” and zygotene”. Move P values from the supplement to the main figure, putting them under each relevant graph.

FigS1: not clear why the D values as defined were chosen to be provided- this parameter seems rather uninformative for capturing important features of the data distribution.

Figure 2 and related text:

How were foci manually selected (what criteria)? How were the ROIs selected (centered on what)? How was the size of the ROIs chosen?

Fig 2G x-axis label “Number of clusters in ROI”

P9: Any claims about differences in the frequencies of different focus categories should be backed up statistically.

Figure 3 and related text:

P9-10: Again, claims are made regarding classes that increase during meiotic progression in WT or in the Sycp1 mutant; need to show statistical support to make these claims.

Also, this is where it begins to be relevant and important to display the information in two different ways, so that BOTH the FRACTION of foci that are of each given class and the (per nucleus average) NUMBERS of foci of each class at a given stage are represented. This is necessary for thinking reasonably about whether it is appropriate to consider one class transitioning to another as prophase progresses. It may be worth considering stacked bar graphs as a useful way for representing both of these points, representing each category with a different color to make it easier to compare the distributions for each stage side by side. Here, the FRACTION graph uses the “parts of a whole” approach, totaling to 100%, while in the average NUMBERS graph the totals would be different for each stage. This would be especially helpful in thinking about comparisons between WT and Sycp1 mutant, where the numbers of foci present at the different stages are very different.

Figure 4 and related text:

As presented, I found this figure very confusing, in part because initially I wasn’t quite sure what you meant by “absolute frequency”, “average frequency” and “relative frequency”. I ended up concluding that this was trying to represent the same things I discussed above for Figure 3 (i.e. fraction in each class, vs average number [per nucleus] in each class). This was confusing in part because of the axis labels and in part because only the foci in these three classes were being considered, which may be problematic for thinking about progression, as discussed above.

If the data in Fig 3 are presented in stacked bar graphs as discussed above, it is likely that this figure will be unnecessary, as the information will already be represented in a way that makes stage to stage comparisons easier and more intuitive.

The only piece of information that would not be represented already is 4C.

L 235-6: reword (and provide p-value), as the current sentence construction is inaccurate, since all classes of foci occur preferentially on unsynapsed axes (as opposed to synapsed axes).

Figs 6 and 7:

Include legend to indicate the use of color in the quadrant maps.

Fig 8:

The graphs in C are so tiny and so blurry that I didn’t even notice the blue and yellow dots until I read the figure legend. Increase size of graphs, increase relative size of the data dots, and indicate what they represent on the figure itself.

Discussion:

L385: Here should include reference to the C. elegans data on paired RAD51 foci.

L426: “… increase in both the number and fraction of foci in the D1R1 configuration…” . Also in this section, consider how you are using the term frequency, as discussed above.

Methods:

Nearest neighbor: what fraction of foci were outside the mask? What was the width of the mask?

Random simulation: see above regarding Fig 1

**Have all data underlying the figures and results presented in the manuscript been provided?**

Reviewer #1: Yes

PLOS authors have the option to publish the peer review history of their article (what does this mean?). If published, this will include your full peer review and any attached files.

Reviewer #1: No

---

## [Decision Letter · Decision Letter 1]

7 Apr 2020

Dear Dr. Baarends,

Thank you very much for submitting your Research Article entitled 'Super-resolution imaging of RAD51 and DMC1 in DNA repair foci reveals dynamic distribution patterns in meiotic prophase' to PLOS Genetics. Your manuscript was fully evaluated at the editorial level and by an independent peer reviewer. The reviewer and I appreciated your revisions and I find the paper is now nearly acceptable.  There are, however, a few more points that I'd like you to address, in addition to those provided by the reviewer.  

Line126 “selected foci in leptotene and zygotene” do you mean “selected nuclei”?

Line 156 Why did you not consider the possibility that the non-random peak at 800 nm represents the equivalent of the paired co-foci we observed at around 400 nm in budding yeast? 

Line 251 Don’t you need to specify "paired" co-foci? i.e. “confirmed the notion that **paired **RAD51-DMC1 co(nano)foci are rare”  

Line 311-314. I think this passage is still unclear. I think you are trying to say something like “the greater distance between the two centers of mass of RAD51-RAD51 (or DMC1-DMC1) nano-focus pairs, as compared to the distance between the two centers of mass of adjacent Rad51-Dmc1 pairs, truly reflects greater spatial separation and cannot be accounted for by elongated focus shape alone.”

Line 463.  See comment re Line 156. You do seem to entertain the possibility that the two ends could be farther apart than 300 nm, but why then is the 800 nm peak not a candidate for truly paired co-foci? 

Line 535-539. We proposed that more than one focus could occupy a ssDNA tract on one end of a DSB given the small size of foci relative to the predicted length of ssDNA tracts. Are you excluding this possibility for some reason?

Fig 2F.  I apologize that I did not examine the high-resolution version of this figure as carefully as I might have on the first submission.  Now that I am looking carefully at the high res version of the figure, I see what you were talking about regarding “clusters.” For example, many D1R1s have multiple distinct maxima for at least one of the two immuno-stains. I appreciate that the approach of converting to binary images was needed to reduce the complexity of the images to a form that allowed you to place ROIs into categories, but clearly a lot of detail was lost as a result of the conversion. I think a few sentences should be added to the discussion to highlight this issue—i.e. that the method of analysis simplified the structural complexity of ROIs significantly, but the simplification was required to provide a means to detect differences in focus configurations between stages and genotypes. 

Also, I think the way Fig 2F is designed is awkward. Rather than having a color-coded boundary on different parts of the figure, it would be simpler to separate the groups of examples with a white border and place a label above each group.  

What are the examples below the blue border that have no designation currently?  

Same issues apply for Sup Figure 3 E. 

Fig 6. The use of standard error in the plot in C obscures the large range of size measurements represented in B. I wonder if a scatter plot or some other method of data presentation would be more informative regarding the obviously wide range of axis-to-focus-center measurements. 

Sup Fig 2C. Eliminate the redundant presentation of the data resulting from the symmetry of the table format. Boxes on and above the diagonal should be eliminated as should the P column.  Same change for Sup Figure 3C; there are only 3 comparisons in that table.

Yours sincerely,

Douglas K. Bishop

Guest Editor

PLOS Genetics

Gregory Barsh

Editor-in-Chief

PLOS Genetics

Reviewer's Responses to Questions

**Comments to the Authors:**

Reviewer #1: I am satisfied with the authors’ responses, which have made the figures and the descriptions of the analyses more clear.

The new analysis measuring the distances of foci to the axis is a nice addition to the work- here it would be more informative to represent the range of distance measurements rather than a simple bar graph with SEM error bars – at a minimum, use SD for error bars, but best to use either box and whiskers or vertical scatter plot so the full distribution and range of distances can be visualized.

The more complete description of the method for generating random distributions for Figs 1 and S1 is appreciated. I also like the solution the authors chose to address the reasons for and possible complications caused by using a mask, i.e., by presenting nearest-neighbor analyses conducted both with and without an Sycp3 mask. The only remaining comments I have are in regards to the presentation and outcomes of these analyses:

In the legend for Fig 1, please state that the distance distributions are for foci associated with Sycp3 masked regions.

In the legend for Fig S1, please state that the distance distributions are for all foci within the analyzed nuclei.

I am struck by the fact that the sharp peaks for D-D and R-R nearest neighbor distances at 700-800 nm not only persisted when all foci were analyzed, but actually increased (as a fraction of total distances). I appreciate that the authors are being judiciously circumspect about interpreting the possible basis for the 800 nm peaks, but this finding does warrant further discussion. The fact that this feature was not dampened, but instead emphasized in the “all foci” analysis suggests that use of the axis mask was partially obscuring the effect. This in turn raises the possibility that some axis associated foci might have a corresponding partner focus located out in a loop and/or extended along the axis, and/or reaching out to the homolog – perhaps the other side of the DSB? The 700-800 nm distance is clearly longer than the typical distances between the paired foci observed in yeast or worms, but is it really ruled out that they could represent partner foci? At a minimum, this issue needs to be discussed, and may affect how some statements are made in the abstract.

**Have all data underlying the figures and results presented in the manuscript been provided?**

Reviewer #1: Yes

PLOS authors have the option to publish the peer review history of their article (what does this mean?). If published, this will include your full peer review and any attached files.

Reviewer #1: No

---

## [Editor Report · Decision Letter 2]

5 May 2020

Dear Dr Baarends,

We are pleased to inform you that your manuscript entitled "Super-resolution imaging of RAD51 and DMC1 in DNA repair foci reveals dynamic distribution patterns in meiotic prophase" has been editorially accepted for publication in PLOS Genetics. Congratulations!

Yours sincerely,

Douglas K. Bishop

Guest Editor

PLOS Genetics

Gregory Barsh

Editor-in-Chief

PLOS Genetics

Comments from the reviewers (if applicable):

**Data Deposition**

http://datadryad.org/submit?journalID=pgenetics&manu=PGENETICS-D-19-02118R2

**Press Queries**

---

## [Editor Report · Acceptance letter]

29 May 2020

PGENETICS-D-19-02118R2 

Super-resolution imaging of RAD51 and DMC1 in DNA repair foci reveals dynamic distribution patterns in meiotic prophase 

Dear Dr Baarends, 

We are pleased to inform you that your manuscript entitled "Super-resolution imaging of RAD51 and DMC1 in DNA repair foci reveals dynamic distribution patterns in meiotic prophase" has been formally accepted for publication in PLOS Genetics! Your manuscript is now with our production department and you will be notified of the publication date in due course.

With kind regards,

Jason Norris

PLOS Genetics

On behalf of:
